# Squamous trans-differentiation of pancreatic cancer cells promotes stromal inflammation

Tim DD Somerville[1], Giulia Biffi[1,2,3], Juliane Daßler-Plenker[1], Stella K Hur[1], Xue-Yan He[1], Krysten E Vance[4], Koji Miyabayashi[1,2], Yali Xu[1], Diogo Maia-Silva[1,5], Olaf Klingbeil[1], Osama E Demerdash[1], Jonathan B Preall[1], Michael A Hollingsworth[4], Mikala Egeblad[1], David A Tuveson[1,2], Christopher R Vakoc[1]*

[1]Cold Spring Harbor Laboratory, Cold Spring Harbor, United States; [2]Lustgarten Foundation Pancreatic Cancer Research Laboratory, Cold Spring Harbor, United States; [3]Cancer Research United Kingdom Cambridge Institute, University of Cambridge, Cambridge, United Kingdom; [4]Eppley Institute for Research in Cancer and Allied Diseases, University of Nebraska Medical Center, Omaha, United States; [5]Watson School of Biological Sciences, Cold Spring Harbor, United States

*For correspondence:
vakoc@cshl.edu

**Abstract** A highly aggressive subset of pancreatic ductal adenocarcinomas undergo trans-differentiation into the squamous lineage during disease progression. Here, we investigated whether squamous trans-differentiation of human and mouse pancreatic cancer cells can influence the phenotype of non-neoplastic cells in the tumor microenvironment. Conditioned media experiments revealed that squamous pancreatic cancer cells secrete factors that recruit neutrophils and convert pancreatic stellate cells into cancer-associated fibroblasts (CAFs) that express inflammatory cytokines at high levels. We use gain- and loss-of-function approaches to show that squamous-subtype pancreatic tumor models become enriched with neutrophils and inflammatory CAFs in a p63-dependent manner. These effects occur, at least in part, through p63-mediated activation of enhancers at pro-inflammatory cytokine loci, which includes *IL1A* and *CXCL1* as key targets. Taken together, our findings reveal enhanced tissue inflammation as a consequence of squamous trans-differentiation in pancreatic cancer, thus highlighting an instructive role of tumor cell lineage in reprogramming the stromal microenvironment.

## Introduction

Pancreatic ductal adenocarcinoma (PDA) is one of the most lethal human tumors, with a five year survival rate below 10% (*Siegel et al., 2020*). Despite the overall association with poor clinical outcomes, a striking heterogeneity in the presentation and progression of this disease exists between individuals, including differing rates of metastatic spread and responses to cytotoxic chemotherapy (*Collisson et al., 2019*). Owing to this issue, a major objective in PDA research is to uncover molecular mechanisms that underpin this clinical heterogeneity. Such efforts may yield much-needed biomarkers capable of predicting disease progression, as well as personalized therapeutic strategies that target critical disease drivers in each patient's tumor.

It has long been recognized that a subset of pancreatic tumors exhibit an 'adenosquamous' histology, which is characterized by the presence of both glandular and squamous neoplastic cells within the same tumor (*Ishikawa et al., 1980*; *Morohoshi et al., 1983*; *Motojima et al., 1992*). The designation of adenosquamous pancreatic cancer as a disease entity has been reinforced by several recent transcriptome profiling studies of human pancreatic tumors, which identified aberrant

expression of squamous (also known as basal) lineage markers in ~15% of samples, in association with exceptionally poor clinical outcomes (*Bailey et al., 2016*; *Cancer Genome Atlas Research Network, 2017*; *Moffitt et al., 2015*). While squamous cells are a normal cell type in several stratified epithelial tissues (e.g. in the skin and esophagus), they are not known to be present in the normal human pancreas (*Basturk et al., 2005*). This suggests an aberrant ductal-to-squamous epithelial cell fate transition is induced during pancreatic tumorigenesis.

The functional relevance of squamous trans-differentiation in pancreatic cancer was unclear until recently, when studies from several laboratories, including our own, demonstrated that the transcription factor p63 (ΔN isoform, hereafter referred to as p63 for simplicity) is the master regulator of the adenosquamous phenotype in PDA (*Andricovich et al., 2018*; *Hamdan and Johnsen, 2018*; *Somerville et al., 2018*). The p63 protein is encoded by the *TP63* gene in humans. Once expressed, p63 binds to thousands of genomic sites to nucleate the formation of active enhancers to drive expression of squamous lineage genes (e.g. *KRT5/6* and *S100A2*) in PDA cells (*Hamdan and Johnsen, 2018*; *Somerville et al., 2018*). Importantly, p63 is both necessary and sufficient to endow PDA cells with the same squamous lineage transcriptional program that is observed in pancreatic tumors with adenosquamous histology (*Bailey et al., 2016*; *Hamdan and Johnsen, 2018*; *Somerville et al., 2018*). While the mechanisms that induce p63 expression in PDA remain unclear, it has been shown in mice that inactivation of the tumor suppressor *Kdm6a* or over-expression of *Myc* can predispose pancreatic tumors to express squamous markers, albeit with partial penetrance (*Andricovich et al., 2018*; *Hayashi et al., 2020*; *Witkiewicz et al., 2015*). In support of the functional impact of squamous trans-differentiation in PDA, induction of p63 leads to a collection of cell-autonomous phenotypic alterations, including enhanced motility, invasion, and resistance to cytotoxic chemotherapy (*Danilov et al., 2011*; *Somerville et al., 2018*).

Evidence from human patients and mouse models supports a powerful effect of inflammation in driving PDA progression (*Cobo et al., 2018*; *Guerra et al., 2011*; *Guerra et al., 2007*; *Yadav and Lowenfels, 2013*). A key step in this process is an elaboration of pro-inflammatory cytokines by tumor cells and non-neoplastic cells in the stromal compartment (*Mantovani et al., 2008*). For example, many pancreatic tumors are infiltrated with neutrophils, which suppress anti-tumor T-cell immunity in PDA mouse models and correlate with aggressive disease in humans (*Bayne et al., 2012*; *Chao et al., 2016*; *Inoue et al., 2015*; *Shen et al., 2014*; *Steele et al., 2016*). A major cell type in the stroma of PDA tumors are CAFs, which have historically been considered tumor-promoting (*Öhlund et al., 2014*), however recent studies suggest they may also have tumor-restraining functions (*Özdemir et al., 2014*; *Rhim et al., 2014*). A subset of CAFs with myofibroblastic properties, termed myCAFs, appear to be involved in the production of extracellular matrix which may limit drug delivery to the tumor (*Olive et al., 2009*; *Provenzano et al., 2012*; *Sherman et al., 2014*). Another subset of CAFs, termed iCAFs, have low expression of myCAF markers and instead produce high levels of inflammatory cytokines and chemokines, such as IL-6, CXCL1, and LIF (*Öhlund et al., 2017*). Although the iCAF/myCAF ratio, the extent of neutrophil infiltration, and the overall level of stromal inflammation can vary between PDA patients, the underlying features of cancer cells that drive these differences are only beginning to be understood (*Biffi et al., 2019*; *Steins et al., 2020*; *Vennin et al., 2019*).

Here we provide evidence that p63-expressing squamous PDA cells have an enhanced capability to promote inflammatory changes within the tumor microenvironment when compared to p63-negative PDA cells. These findings are supported by in vitro and in vivo experiments, and include activation of an inflammatory transcriptional program in CAFs of the tumor stroma. These effects are mediated via a specific cytokine secretion phenotype of PDA cells downstream of p63-mediated enhancer reprogramming. Taken together, these findings illustrate how lineage alterations within cancer cells orchestrate non cell-autonomous changes in the tumor microenvironment.

## Results

### A secretory phenotype of p63-positive PDA cells that promotes inflammatory gene expression changes in CAFs in vitro

In a prior study, we found that ectopic expression of p63 in the human PDA cell line SUIT2, which lacks endogenous p63 expression, enhanced cell growth when implanted into the mouse pancreas

in vivo, yet this phenotype was absent under tissue culture conditions (*Somerville et al., 2018*). These findings led us to hypothesize that p63 expression alters how PDA cells communicate with non-neoplastic cells in the tumor stroma. We focused our studies initially on CAFs, which have established roles in PDA pathogenesis (*Pereira et al., 2019*). To this end, we collected conditioned media from 12 human PDA cell lines exhibiting different expression levels of squamous markers (e.g. *TP63* and *KRT5*) (*Figure 1A–C* and *Figure 1—figure supplement 1A*). Of note, MIAPaca2 cells only express the TA isoform of p63 (*Figure 1—figure supplement 1A*) and lack expression of squamous lineage markers (*Somerville et al., 2018*), and in fact are more representative of a cell type of neuro-endocrine origin (*Yu et al., 2019*). The collected media were then applied to quiescent murine pancreatic stellate cells (PSCs), which are a precursor of CAFs in PDA, and RNA-sequencing (RNA-seq) analysis was performed on the PSC cultures following 96 hr to evaluate the cellular response (*Figure 1D*). An unsupervised clustering of the global transcriptional profile of the PSCs revealed three distinct groups (*Figure 1E*). Two of the treated PSC cultures clustered closely with the untreated controls, suggesting they remained in a quiescent transcriptional state (*Figure 1E*). However, two major groups were found to cluster away from the control PSC cultures and were termed Group 1 and Group 2 (*Figure 1E*). Remarkably, the PSCs within the Group 2 cluster were all treated with conditioned media derived from the three PDA cell lines expressing p63 (BxPC3, T3M4, and KLM1; *Figure 1A–B* and *Figure 1—figure supplement 1A*). In addition to this transcriptional phenotype, the conditioned media from p63-expressing lines also led to a stronger induction of PSC proliferation than conditioned media collected from p63-negative lines (*Figure 1—figure supplement 1B–C*). We next extracted the subset of genes that discriminate the Group 1 and Group 2 clusters of PSC transcriptional responses, identifying 144 and 259 differentially expressed genes, respectively (*Figure 1F* and *Supplementary file 1*). A Gene Set Enrichment Analysis (GSEA) of the Group 2 cluster revealed several top-ranking gene signatures associated with inflammation (*Figure 1G* and *Figure 1—figure supplement 1D*). Using transcriptome data from our prior study in which we identified iCAFs and myCAFs (*Öhlund et al., 2017*), we defined gene signatures associated with these two cell fates (*Supplementary file 2*) and observed a significant enrichment of the iCAF signature within the Group 2 set of PSCs (*Figure 1H*). In contrast, the myCAF signature was significantly enriched within Group 1 PSCs (*Figure 1I*). RT-qPCR analysis of human PSCs treated with conditioned media from PDA cell lines further validated these findings (*Figure 1—figure supplement 1E*). These findings suggested that a pro-inflammatory secretory phenotype is linked with the acquisition of squamous features in PDA.

## p63 expression in PDA cells drives a secretory phenotype that induces iCAF formation in vitro

Based on the correlations described above, we next set out to determine the causality between p63 and the iCAF-inducing secretory phenotype. To this end, we cultured two mouse PSC lines with conditioned media harvested from SUIT2-empty and SUIT2-p63 cells and performed RT-qPCR analysis of iCAF marker genes in the PSC cultures (*Figure 2A*). We observed a marked increase in iCAF markers (*Il6*, *Cxcl1*, and *Lif*) and a concomitant reduction of myCAF markers (*Acta2* and *Ctgf*) when the PSCs were cultured in the SUIT2-p63 conditioned media compared to those cultured in SUIT2-empty conditioned media (*Figure 2B–C* and *Figure 2—figure supplement 1A–B*). In accord with the findings described above, mouse PSCs cultured in the SUIT2-p63 conditioned media were more proliferative when compared to their counterparts cultured in SUIT2-empty conditioned media (*Figure 2—figure supplement 1C–D*). Similar RT-qPCR experiments using human PSC cultures or using a metastatic mM1 organoid line, which was isolated from the KPC (*Kras*$^{+/LSL-G12D}$; *Trp53*$^{+/LSL-R172H}$; *Pdx1-Cre*) mouse model (*Boj et al., 2015*; *Hingorani et al., 2005*), and engineered to ectopically express p63, support the generality of these findings (*Figure 2D–E* and *Figure 2—figure supplement 1E–F*).

We next performed loss-of-function studies using conditioned media from KLM1-Cas9 cells infected with two independent sgRNAs targeting *TP63* or a control sgRNA (*Figure 2F*). Notably, KLM1 is the only p63-positive PDA cell line we have identified which does not exhibit a growth arrest phenotype in vitro following p63 inactivation (*Figure 2—figure supplement 1G–H*; *Somerville et al., 2018*). Importantly, KLM1 cells harbor a similar transcriptional and epigenomic profile as observed in other p63-expressing PDA cell lines (*Figure 2—figure supplement 1I–K*). In addition, KLM1 cells also induce a similar iCAF phenotype as seen with other p63-expressing lines

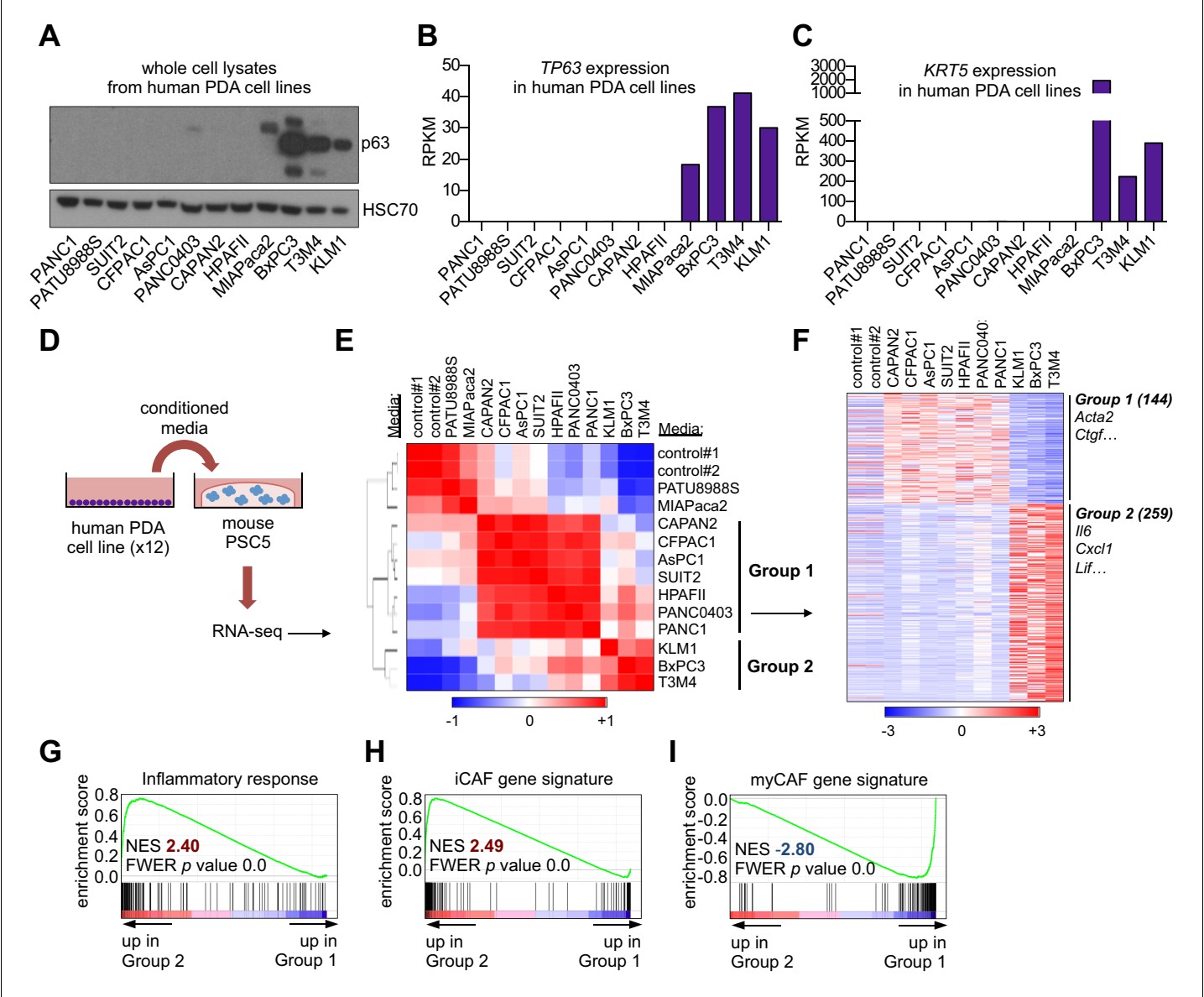

**Figure 1.** A secretory phenotype of p63-positive PDA cells that promotes inflammatory gene expression changes in CAFs in vitro. (A) Western blot analysis showing p63 expression in a panel of 12 human PDA cell lines. (B–C) Bar charts show expression of *TP63* and *KRT5* cell lines shown in (A). (D) Schematic of experimental workflow for RNA-seq analysis of PSCs following culture in Matrigel with conditioned media from the 12 human PDA cell lines. (E) Heatmap representation of unsupervised hierarchical clustering of mouse PSCs based on their global transcriptional profile. Scale bar indicates Pearson correlation coefficient. Control refers to control media, which was DMEM supplemented with 5% FBS. (F) Heatmap representation of differentially expressed genes from PSCs in Group 1 and Group 2. Selected genes in each group are listed. Scale bar indicates standardized expression value. (G) GSEA plot evaluating Hallmark Inflammatory Response genes based on their expression in Group 2 versus Group 1 cultures. (H–I) GSEA plots evaluating the iCAF and myCAF gene signatures in Group 2 versus Group 1 cultures.

The online version of this article includes the following figure supplement(s) for figure 1:

**Figure supplement 1.** A secretory phenotype of p63-positive PDA cells that promotes inflammatory gene expression changes in CAFs in vitro.

(*Figure 1E–F*). Hence, this cell line is useful for performing p63 loss-of-function experiments evaluating for secretory phenotypes without the confounding effect of impairing cell fitness.

While conditioned media from KLM1 cells led to a significant induction of iCAF markers and repression of myCAF markers in mouse and human PSCs, this effect was significantly attenuated in p63-knockout KLM1 cells (*Figure 2G–I* and *Figure 2—figure supplement 1L–N*). In addition, the proliferation phenotype of the conditioned media-treated PSCs was also attenuated following p63

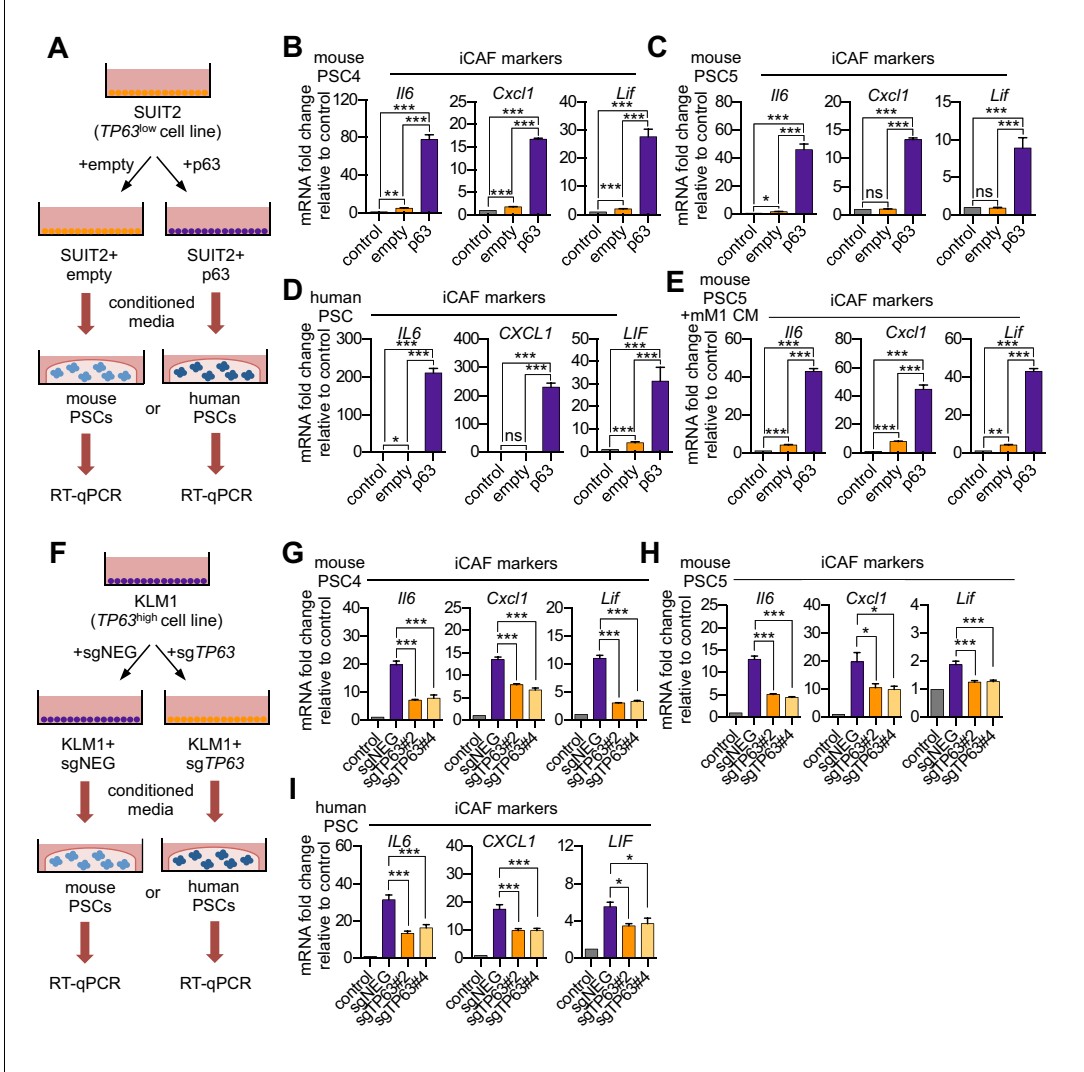

**Figure 2.** p63 expression in PDA cells drives a secretory phenotype that induces iCAF formation in vitro. (A) Schematic of workflow for generating conditioned media from SUIT2-empty and SUIT2-p63 cells and subsequent analysis of PSC cultures. (B–C) Bar charts showing RT-qPCR analysis for iCAF markers (*Il6*, *Cxcl1*, *Lif*) following culture of 2 mouse PSC lines in Matrigel for four days with the indicated conditioned media. (D) RT-qPCR analysis for the iCAF markers (*IL6*, *CXCL1*, *LIF*) following culture of human PSCs in Matrigel for four days with the indicated conditioned media. (E) Mouse mM1 organoids were infected with a p63 cDNA or empty vector control. Bar chart shows RT-qPCR analysis for iCAF markers (*Il6*, *Cxcl1*, *Lif*) following culture of a mouse PSC line in Matrigel for four days in the indicated conditioned medias. (F) Schematic of workflow for generating conditioned media from KLM1-Cas9 cells infected with p63 sgRNAs (sg*TP63*#2 and sg*TP63*#4) or a control (sgNEG) and subsequent analysis of PSC cultures. (G–H) Bar chart showing RT-qPCR analysis for iCAF markers (*Il6*, *Cxcl1*, *Lif*) following culture of mouse PSCs in Matrigel for four days with the indicated conditioned media. (I) RT-qPCR analysis for iCAF markers (*IL6*, *CXCL1*, *LIF*) following culture of human PSCs in Matrigel for four days with the indicated conditioned media. For all experiments, control media represents DMEM supplemented with 5% FBS. Mean+ SEM is shown. Results are presented as the mean of three technical replicates and are representative of n = 3 biological replicates. For B-E, ***p<0.001, **p<0.01, *p<0.05 by Student's t-test and for G-I, ***p<0.001, **p<0.01, *p<0.05 by one-way ANOVA with Dunnett's test for multiple comparisons.

The online version of this article includes the following figure supplement(s) for figure 2:

**Figure supplement 1.** p63 expression in PDA cells drives a secretory phenotype that induces iCAF formation in vitro.

inactivation (*Figure 2—figure supplement 1O–P*). Taken together, these experiments demonstrate the critical role of p63 in regulating the secretory phenotype in squamous-like models of PDA, which promotes the induction of inflammation-associated transcriptional changes in CAFs.

## Ectopic expression of p63 in PDA cells promotes inflammation-associated transcriptional changes in the tumor microenvironment in vivo

We next evaluated the relevance of the p63-induced secretory phenotype in an in vivo model of PDA. For this purpose, we first infected SUIT2-luciferase cells (*Somerville et al., 2018*) with a p63 cDNA or empty vector prior to orthotopic transplantation into the pancreas of NOD-scid gamma (NSG) mice. In accordance with our previous findings (*Somerville et al., 2018*), p63 expression enhanced the growth of SUIT2 cells in vivo, but not in vitro (*Figure 3A–B* and *Figure 3—figure supplement 1A–D*). Importantly, we validated the in vivo-specific growth advantage caused by ectopic p63 expression in the mM1 organoids following transplantation into the pancreas of C57BL/6 mice (*Figure 3—figure supplement 1E–K*). Tumors from the SUIT2 xenograft model were harvested and subjected to bulk RNA-seq analysis. Mapping of human and mouse transcripts to their respective genomes allowed us to discriminate the transcriptional changes happening in the human cancer cells from those of non-neoplastic mouse cells comprising the tumor stroma (*Figure 3C*).

In the human cancer cell compartment, we identified 633 genes that were significantly upregulated in SUIT2-p63 tumors compared to controls, which largely corresponded to squamous lineage p63 target genes (e.g. *S100A2*, *KRT5*) (*Figure 3D*, *Figure 3—figure supplement 1L* and *Supplementary file 3*; *Bailey et al., 2016*; *Somerville et al., 2018*). Within the mouse stromal compartment, we identified 500 genes that were significantly upregulated in SUIT2-p63 tumors compared to controls (*Figure 3E* and *Supplementary file 3*). An ontology analysis of this set of genes revealed enrichment for inflammatory responses, neutrophil degranulation, and cytokine production as top-ranking gene sets (*Figure 3F*). Importantly, the transcriptional signature of iCAFs was significantly more enriched in the mouse stromal compartment of SUIT2-p63 relative to SUIT2-empty tumors (*Figure 3G*), which is consistent with our conditioned media experiments (*Figure 2A–D*). In this bulk RNA-seq analysis, the transcriptional signature of myCAFs was modestly depleted in SUIT2-p63 tumors, although this was not significant (*Figure 3—figure supplement 1M*). However, fluorescent activated cell sorting (FACS) of fibroblasts from these two groups of tumors confirmed an increase in iCAF markers and a decrease in myCAF markers in the SUIT2-p63 tumors (*Figure 3—figure supplement 1N–P*). Flow cytometry analysis confirmed that SUIT2-p63 tumors were more infiltrated with $CD45^+CD11b^+Ly6G^+$ neutrophils when compared to control tumors, which is a marker of enhanced tissue inflammation (*Figure 3H–I*). iCAF induction and neutrophil infiltration were also correlated with pre-existing levels of *TP63* expression in human PDA organoid xenografts models (*Figure 3—figure supplement 1Q–T*). Moreover, a similar neutrophil infiltration phenotype was observed in the orthotopic mM1-p63 tumors, which also correlated with a decrease in infiltrating CD4- and CD8-positive T cells (*Figure 3J–K*). This finding is consistent with prior evidence that tumor-associated neutrophils have an immunosuppressive effect in PDA (*Bayne et al., 2012*; *Chao et al., 2016*). Finally, plasma concentrations of the murine cytokine IL-6, a marker of systemic inflammation and immunosuppression (*Flint et al., 2016*; *Nishimoto and Kishimoto, 2006*) that is largely produced by iCAFs in PDA (*Biffi et al., 2019*; *Elyada et al., 2019*; *Öhlund et al., 2017*), were ~10 fold higher in the mice bearing SUIT2-p63 versus SUIT2-empty tumors (*Figure 3—figure supplement 1U*). Collectively, these experiments suggest that p63-positive PDA cells induce enhanced stromal inflammation in vivo when compared to p63-negative tumors.

## Knockout of p63 in an orthotopic PDA tumor model attenuates stromal inflammation

We next addressed the impact of p63 inactivation on the tumor microenvironment in vivo. To this end, KLM1-Cas9-luciferase cells were transduced with *TP63* sgRNAs before transplantation into the pancreas of NSG mice (*Figure 4—figure supplement 1A*). By monitoring tumor progression using bioluminescent imaging, we found that p63 inactivation resulted in a significant reduction in tumor growth when compared to control tumors (*Figure 4A–B*), which is consistent with in vivo-specific growth enhancement caused by acute p63 expression in SUIT2 and mM1 orthotopic models (*Figure 3A–B* and *Figure 3—figure supplement 1A–K*). In order to study the impact of p63 inactivation on the tumor microenvironment while controlling for overall tumor burden, we allowed sufficient time for tumors to form in each group before harvesting for analysis when a critical bioluminescent signal was reached. By this endpoint metric, cells transduced with the two independent *TP63*

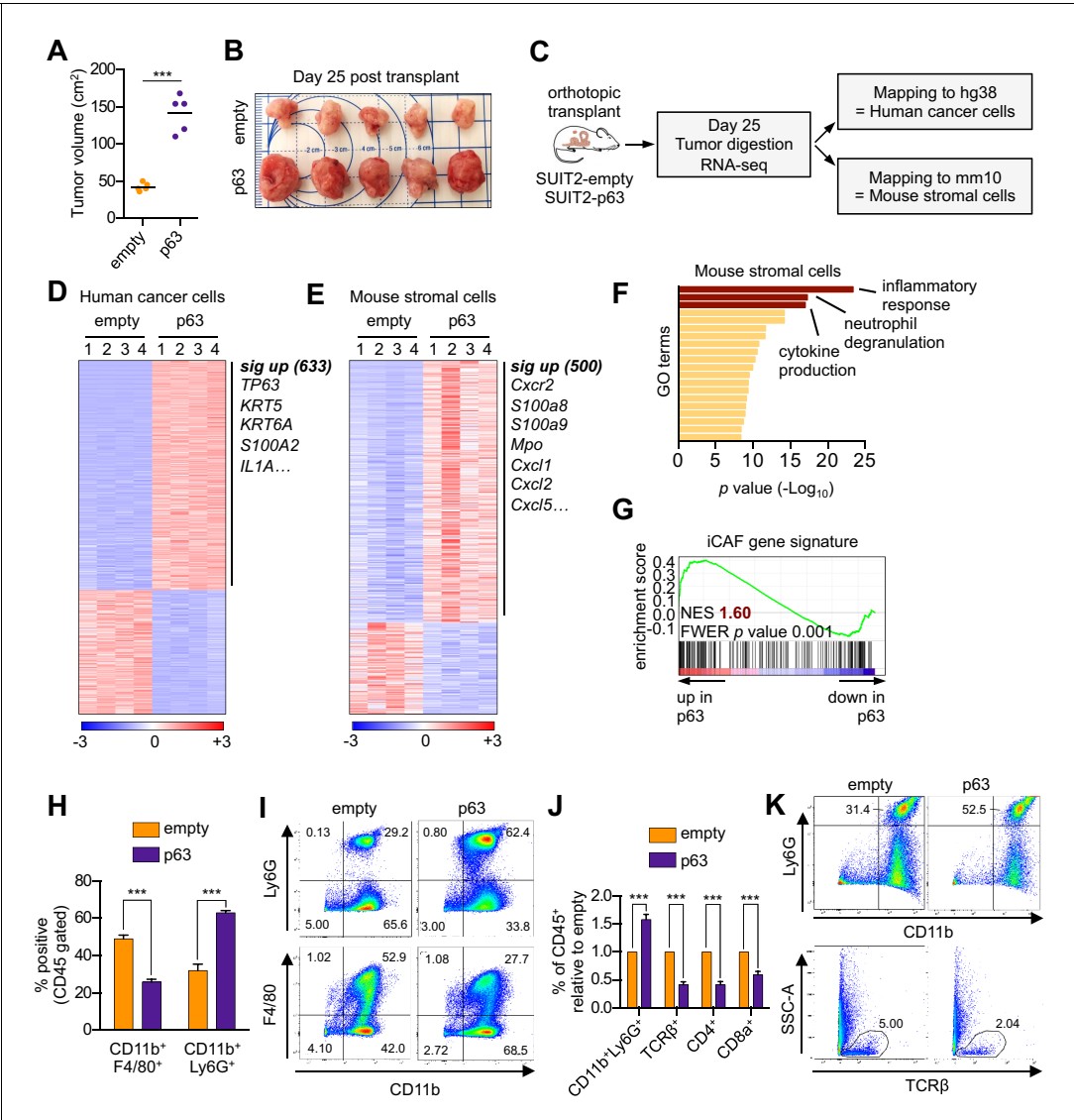

**Figure 3.** Ectopic expression of p63 in PDA cells promotes inflammation-associated transcriptional changes in the tumor microenvironment. (A–I) SUIT2-empty and SUIT2-p63 PDA cells harboring a luciferase transgene were transplanted into the pancreas of NSG mice. (A–B) Quantification of tumor volume (A) and images of tumors (B) on day 25 post-transplantation. (C) Experimental workflow for RNA-seq analysis of bulk tumor tissue. (D–E) Heatmap representations of differentially expressed genes that unambiguously map to (D) the human genome and therefore are derived from human cancer cells or (E) the mouse genome and therefore are derived from mouse stromal cells. Selected genes in each group are listed. Scale bar indicates standardized expression value. (F) Ontology analysis of the 500 significantly up-regulated mouse genes using Metascape. GO terms are ranked by their significance ($p$ value) and the most significant terms (-$\log_{10}$ $p$ value > 15) are highlighted. (G) GSEA plot evaluating the iCAF signature in the mouse stromal compartment of SUIT2-p63 versus SUIT2-empty tumors. (H) Quantification of flow cytometry analysis of CD45$^+$CD11b$^+$Ly6G$^+$ neutrophils and CD45$^+$CD11b$^+$F4/80$^+$ macrophages from bulk tumor tissues from SUIT2 xenografts. n = 5 mice per group. ***$p<0.001$ by Student's t-test. (I) Representative flow cytometry plots from (H). (J–K) KPC mM1 organoids with a p63 cDNA or empty vector control and transplanted to the pancreas of C57BL6 mice. (J) Quantification of flow cytometry analysis of CD45$^+$ cells in the tumors. n = 4–5 mice per group. ***$p<0.001$ by Student's t-test. (K) Representative flow cytometry plots from (J). TCRβ refers to T cell receptor β.

The online version of this article includes the following figure supplement(s) for figure 3:

**Figure supplement 1.** Ectopic expression of p63 in PDA cells promotes inflammation-associated transcriptional changes in the tumor microenvironment.

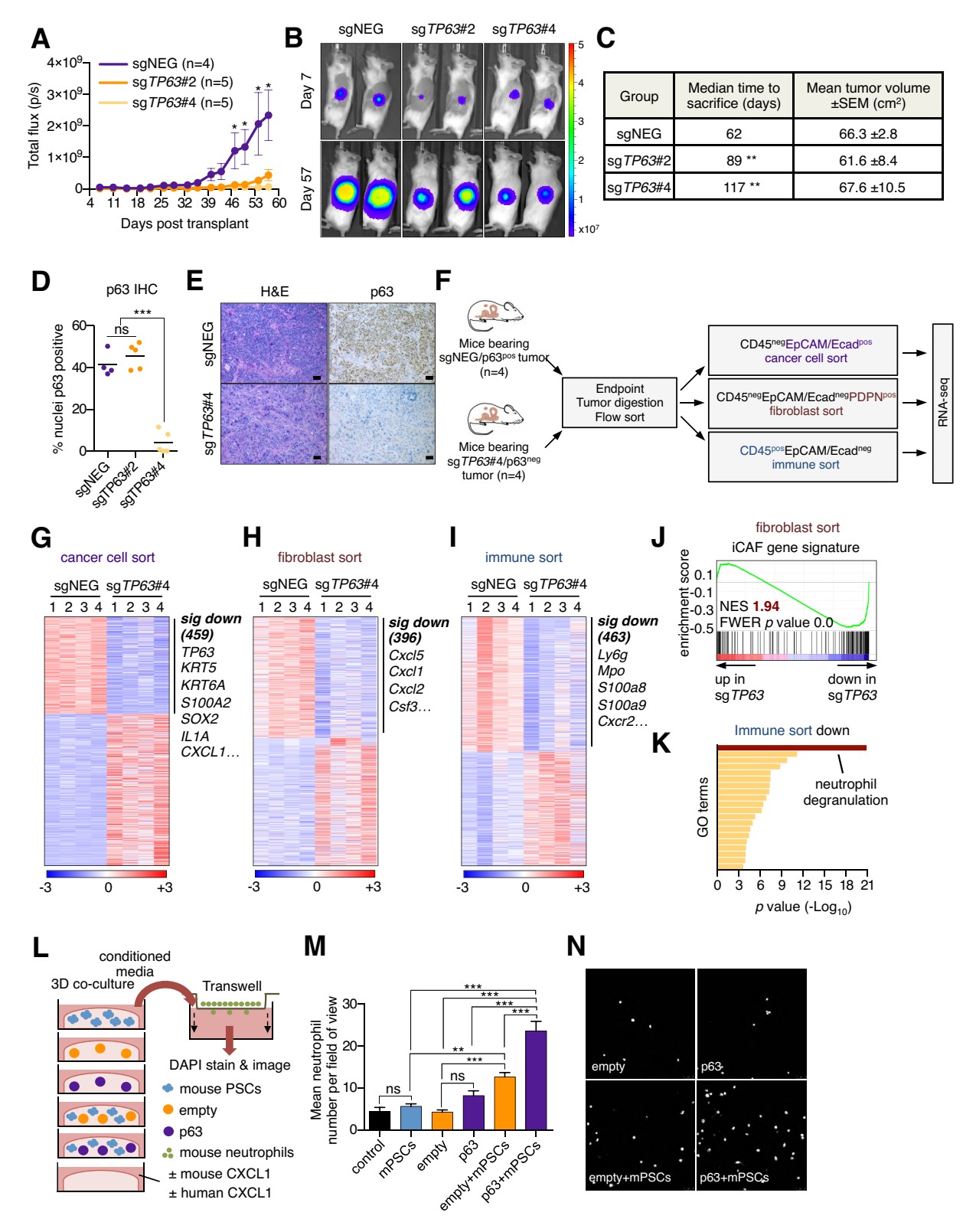

**Figure 4.** Knockout of p63 in an orthotopic PDA tumor model attenuates stromal inflammation. (A–E) KLM1-Cas9 cells expressing a luciferase transgene were infected with two independent p63 or control (sgNEG) sgRNAs and transplanted to the pancreas of NSG mice. (A) Quantification of bioluminescence signal. Mean ± SEM is shown. Mice were imaged every four days between day 7 and day 57 post-transplantation. n = 4–5 mice per group. *p<0.05 by two-way ANOVA with Sidak's test for multiple comparisons. (B) Representative images at day 7 and day 57 post-transplantation from

*Figure 4 continued on next page*

Figure 4 continued

(A). (C) Table summarizing the median time mice from each group were sacrificed and the tumor volume at endpoint. Endpoint was determined as a bioluminescence signal >3×$10^9$ p/s for each individual mouse. **p<0.01 by log rank (Mantel-Cox) test. (D) Quantification of p63 expression as determined by immunohistochemical staining of tumors from the indicated experimental groups. n = 4–5 mice per group. ***p<0.001 by one-way ANOVA with Tukey's test for multiple comparisons. (E) Representative images from (D). Scale bar indicates 50 μm. (F) Schematic of experimental workflow and sorting strategy for enriching for human cancer cells, mouse fibroblasts and mouse immune cells and subsequent RNA-seq analysis. (G–I) Heatmap representations of differentially expressed genes from the human cancer cell compartment (G), mouse fibroblast compartment (H) and mouse immune compartment (I) and the indicated tumor samples. Selected genes in each group are listed. Scale bar indicates standardized expression value. (J) GSEA plot evaluating the iCAF signature in the mouse fibroblast compartment of p63-positive versus control (sgNEG) tumors. (K) Ontology analysis of the 463 significantly down-regulated mouse genes from immune compartment using Metascape. GO terms are ranked by their significance (p value) and the most significant term (-log$_{10}$ p value > 15) is highlighted. (L–N) SUIT2-empty or SUIT2-p63 cells were cultured in isolation or together with mouse PSCs in Matrigel for four days before conditioned media was harvested and used for a trans-well migration assay with freshly isolated mouse neutrophils. (L) Schematic of experimental workflow. (M) Quantification of trans-well neutrophil migration in the indicated media conditions as determined by confocal imaging. Mean+ SEM is shown. Results are presented as the mean number of invading neutrophils per 10 random fields for each condition and are representative of n = 3 biological replicates. ***p<0.001, **p<0.01 by two-way ANOVA with Sidak's test for multiple comparisons. All significant interactions are shown. (N) Representative confocal images from (M).

The online version of this article includes the following figure supplement(s) for figure 4:

**Figure supplement 1.** Knockout of p63 in an orthotopic PDA tumor model attenuates the inflammatory signature of fibroblasts and immune cells.

---

sgRNAs formed tumors with a significantly longer latency versus controls (*Figure 4C*). Immunohisto-chemical analysis for p63 revealed an 'escaper' group (sg*TP63*#2) that gave rise to p63-positive tumors, suggesting the emergence of p63-positive clones within the pooled population of trans-planted cells (*Figure 4D*). However, in the second cohort where p63 was inactivated (sg*TP63*#4), we identified largely p63-negative tumors at endpoint, allowing us to interrogate the tumor microenvi-ronment in a loss of function context (*Figure 4D–E* and *Figure 4—figure supplement 1B*).

To achieve this, we FACS-purified human cancer cells, immune cells, and fibroblasts and per-formed RNA-seq analysis (*Figure 4F–I*). Within the cancer cell compartment, we identified 459 genes that were significantly downregulated in the p63-knockout tumors (*Figure 4G* and *Supplementary file 4*). As expected, this included *TP63* as well as its downstream target genes, *KRT5*, *KRT6A* and *S100A2* (*Figure 4G* and *Figure 4—figure supplement 1C*). Within the sorted fibroblast compartment, we identified 396 genes that were significantly downregulated in the p63-knockout tumors, which included a significant suppression of the iCAF gene signature (*Figure 4H–J* and *Supplementary file 4*). In contrast, the myCAF gene signature was significantly induced in p63-knockout tumors (*Figure 4—figure supplement 1D*). Finally, within the sorted immune cell fraction we identified 463 genes that were significantly down-regulated in the p63-knockout tumors, which was enriched for markers of neutrophils, such as *Ly6g*, *Mpo*, *S100a8*, *S100a9* and *Cxcr2* (*Figure 4I–K* and *Supplementary file 4*). We confirmed reduced neutrophil infiltration in the p63 knockout tumors by immunofluorescence staining for LY6G and myeloperoxidase (MPO) (*Figure 4—figure supple-ment 1E–H*). Taken together, these findings demonstrate that inactivation of p63 dampens the inflammation-associated transcriptional response present in the CAF and immune compartments of the stroma.

The findings above led us to investigate whether p63-expressing PDA cells can collaborate with CAFs to increase neutrophil infiltration. In order to model this in vitro, we established a co-culture system in which human SUIT2-empty or SUIT2-p63 cells were cultured in Matrigel either alone or in combination with mouse PSCs (*Figure 4L*). RT-qPCR analysis for murine-specific transcripts con-firmed that SUIT2-p63 cells augmented the iCAF phenotype versus controls in this co-culture system (*Figure 4—figure supplement 1I*). Of note, this included a marked upregulation of the iCAF gene *Cxcl1*, which encodes an established neutrophil chemoattractant (*Moser et al., 1990*). We found that the SUIT2-p63/PSC co-cultures released soluble factors that increased the chemotaxis of pri-mary mouse neutrophils and, importantly, this phenotype was enhanced in the SUIT2-p63/PSC co-culture as compared to each cell type individually (*Figure 4M–N* and *Figure 4—figure supplement 1J*). Of note, we found that conditioned media from p63-expressing PDA cells could also promote a modest degree of neutrophil recruitment in the absence of co-cultured PSCs (*Figure 4—figure sup-plement 1K-M*). This effect was dependent on CXCL1 (*Figure 4—figure supplement 1K-M*), which is a p63 target gene in PDA cells (see below). Taken together, these experiments suggest that p63-

positive PDA cells possess a secretory phenotype that can promote neutrophil migration, which can be further enhanced by the presence of PSCs/iCAFs.

## p63 activates enhancer elements and transcription of genes encoding pro-inflammatory cytokines in PDA cells

We next sought to define a molecular mechanism by which p63 drives a pro-inflammatory secretory phenotype by interrogating the transcriptional changes that occur following ectopic p63 expression in SUIT2 cells (*Somerville et al., 2018*). GSEA comparing SUIT2-p63 to SUIT2-empty cells revealed the activation of a number of inflammation-related pathways as top-ranking gene sets following p63 expression (*Figure 5A–C* and *Figure 5—figure supplement 1A*). Of note, a number of these pathways (e.g. NF-κB and JAK/STAT3 signaling) overlapped with those activated in iCAFs (*Biffi et al., 2019*). Next, we interrogated the transcriptional changes of inflammatory genes that occur following acute p63 inactivation in BxPC3 cells, a human PDA cell line that expresses p63 at high levels (*Somerville et al., 2018*). These analyses identified *IL1A*, *IL1B*, *CXCL1*, *CXCL8*, *CSF2* (*GM-CSF*), *CCL20*, *AREG*, *FJX1*, and *ADORA2B* as the top inflammation-related genes regulated by p63 (*Figure 5C* and *Figure 5—figure supplement 1B*). Importantly, an analysis of p63 over-expression and knockout/knockdown RNA-seq datasets performed in nine different human PDA lines or organoids (*Somerville et al., 2018*) revealed that this group of inflammatory genes is regulated by p63 across these different contexts, albeit with a degree of heterogeneity (*Figure 5D* and *Figure 5—figure supplement 1C*). Chromatin immunoprecipitation followed by DNA sequencing (ChIP-seq) analysis performed in BxPC3 cells revealed that each of these genes is located near a p63-bound enhancer element, suggesting direct transcriptional regulation (*Figure 5—figure supplement 1D*). Among these p63 targets, *IL1A* and *IL1B* were outliers, as their expression was the most affected by p63 perturbation (*Figure 5D*). ChIP-seq analysis revealed a p63-induced super-enhancer located at an intergenic region between the *IL1A* and *IL1B* genes (*Figure 5E* and *Figure 5—figure supplement 1E–H*). Using CRISPR-interference to target this region for inactivation (*Qi et al., 2013*), we verified that this group of p63-bound enhancers was responsible for activating *IL1A* and *IL1B* expression (*Figure 5E–F* and *Figure 5—figure supplement 1I*). Taken together, these findings suggest that p63 promotes enhancer-mediated transcriptional activation of genes encoding secreted factors with established roles in promoting inflammation.

## IL-1α is the key secreted molecule induced by p63 in PDA cells that promotes iCAF induction in vitro

We focused our subsequent functional experiments on *IL1A* and *IL1B*, which respectively encode interleukin one alpha (IL-1α) and beta (IL-1β) and are important mediators of inflammatory responses, including NF-κB signaling that we found to be upregulated following p63 expression (*Figure 5B* and *Figure 5—figure supplement 1A*; *Di Paolo and Shayakhmetov, 2016*; *Dinarello and Wolff, 1993*). We recently demonstrated a critical role for IL-1 in promoting the induction of iCAFs through the indirect activation of JAK/STAT signaling (*Biffi et al., 2019*). In order to assess whether IL-1α and IL-1β are secreted by p63-expressing PDA cells, we performed enzyme-linked immunosorbent assay (ELISA) experiments for these two proteins on conditioned media following p63 knockout in KLM1 cells. This assay efficiently detected IL-1α, but not IL-1β, in the control setting and the levels of secreted IL-1α were significantly reduced by two independent *TP63* sgRNAs (*Figure 5G*). An absence of IL-1β secretion in this context might be due to a lack of inflammasome activation, which is a known requirement for IL-β, but not IL-α secretion (*Di Paolo and Shayakhmetov, 2016*). Consistent with this observation, ectopic expression of p63 in SUIT2 cells led to robust secretion of IL-1α, but not IL-1β, and this effect could be ablated by CRISPR-interference-mediated targeting of the *IL1* super-enhancer (*Figure 5H–I*). Taken together, these results indicate that both *IL1A* and *IL1B* are regulated by p63 at the transcriptional level, but only IL-1α is secreted at detectable levels by p63-expressing PDA cells in vitro.

In order to investigate whether p63-driven iCAF induction is conferred via an IL-1-mediated signaling mechanism, we used mouse PSC lines in which the IL-1 receptor (*Il1r1*) was knocked out using CRISPR (*Biffi et al., 2019*) and performed experiments with conditioned media harvested from SUIT2-p63 or SUIT2-empty cells. Consistent with the results above, PSCs infected with the control sgRNA displayed a marked increase in iCAF gene expression following culture in SUIT2-p63

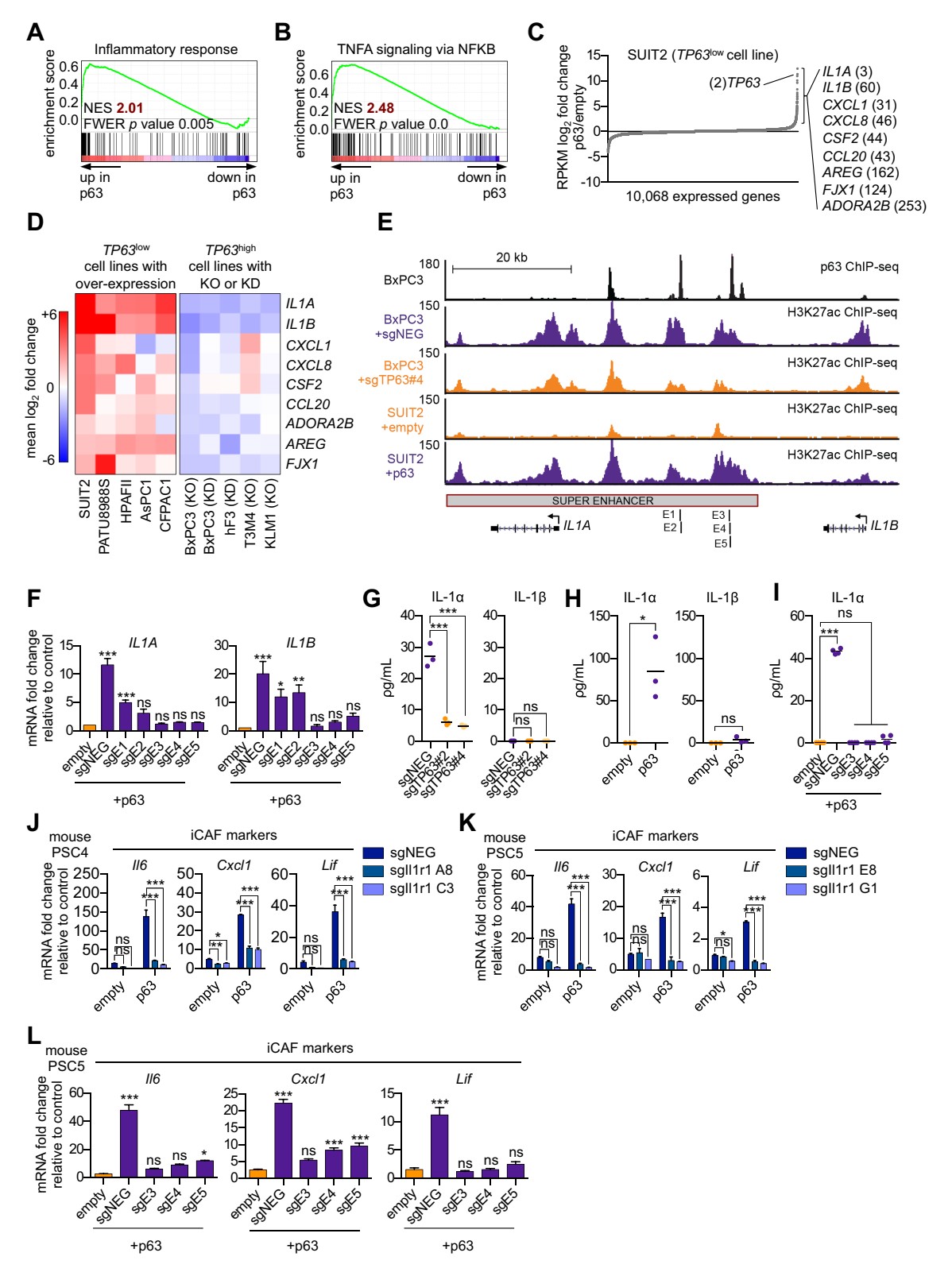

**Figure 5.** p63 activates enhancer elements and transcription of genes encoding pro-inflammatory cytokines in PDA cells. (**A–B**) GSEA plots evaluating the indicated Hallmark gene signatures upon p63 expression in SUIT2 cells. (**C**) Scatter plot shows the mean log₂ fold change of expressed genes upon p63 expression in SUIT2 cells. Genes with a mean log₂ fold change >1 in this dataset and the BxPC3+sg*TP63* dataset (*Figure 5—figure supplement 1B*) that are also found in the gene signatures shown in A and B are highlighted along with the rank. Data are from *Somerville et al. (2018)*. (**D**)

*Figure 5 continued on next page*

Figure 5 continued

Heatmap shows gene expression changes in those genes shown in (C) in the indicated cell lines following p63 over-expression (left panel) or knockout/ knockdown (right panel). Scale bar indicates mean $\log_2$ fold change versus control. (E) ChIP-seq profiles of p63 (top track) and H3K27ac (bottom four tracks) following p63 knockout in BxPC3 cells or overexpression in SUIT2 cells surrounding the *IL1* locus. The H3K27ac regions identified as a super enhancer by ROSE analysis are indicated along with the positions of sgRNAs targeting this enhancer used for experiments shown in (F) and (I). (F) RT-qPCR analysis for *IL1A* and *IL1B* in SUIT2-empty or SUIT2-p63 cells infected with dCas9 fused with the KRAB repression domain and the indicated sgRNAs. The positions of the sgRNAs targeting the *IL1* super enhancer are shown in (E). (G–H) ELISA for IL-1α and IL-1β from conditioned media harvested from KLM1-Cas9 cells infected with the indicated sgRNA (G) or from SUIT2-empty and SUIT2-p63 cells (H). (I) ELISA for IL-1α in SUIT2-empty or SUIT2-p63 cells infected with dCas9 fused with the KRAB repression domain and the indicated sgRNAs. (J–K) Bar charts showing RT-qPCR analysis for iCAF markers (*Il6*, *Cxcl1*, *Lif*) following culture of clones of two mouse PSC lines from which the IL-1 receptor was clonally knocked out with CRISPR in Matrigel for four days with conditioned media from SUIT2-empty or SUIT2-p63 cells. (L) Bar charts showing RT-qPCR analysis for iCAF markers (*Il6*, *Cxcl1*, *Lif*) following culture of PSCs in Matrigel for four days with the conditioned media harvested from cells shown in (I). For all experiments, control media represent DMEM supplemented with 5% FBS. Mean+ SEM is shown. For RT-qPCR experiments, results are presented as the mean of three technical replicates and are representative of n = 3 biological replicates and ***p<0.001, **p<0.01, *p<0.05 by one-way ANOVA with Dunnett's test for multiple comparisons. For each ELISA experiment, n = 3 biological replicates and ***p<0.001, **p<0.01, *p<0.05 by one-way ANOVA with Tukey's test for multiple comparisons. ns = not significant.

The online version of this article includes the following figure supplement(s) for figure 5:

**Figure supplement 1.** p63 activates enhancer elements and transcription of genes encoding pro-inflammatory cytokines in PDA cells.

conditioned media (*Figure 5J–K* and *Figure 5—figure supplement 1J–K*). However, this induction was absent in the IL-1 receptor knock out lines (*Figure 5J–K* and *Figure 5—figure supplement 1J–K*). Finally, we confirmed the dependency on IL-1 signaling for p63 driven iCAF induction by performing conditioned media experiments using SUIT2-p63 cells following inactivation of the *IL1* super-enhancer via CRISPR interference (*Figure 5L* and *Figure 5—figure supplement 1L*). These data demonstrate that p63 expressing PDA cells induce an iCAF phenotype via paracrine IL-1α signaling in vitro.

## Squamous trans-differentiation of human PDA tumors correlates with increased inflammatory fibroblasts and neutrophil infiltration in patient samples

In order to ascertain whether p63 expression is associated with stromal inflammation changes in human pancreatic cancer patients, we first interrogated three independent datasets that have profiled the transcriptome of bulk tumor tissue (*Bailey et al., 2016*; *Cancer Genome Atlas Research Network, 2017*; *Moffitt et al., 2015*). In all three datasets, the expression of *IL1A* was found to be significantly elevated in *TP63*high versus *TP63*low patient samples (*Figure 6A*). Of note, it is the ΔN isoform of *TP63* that is overexpressed in squamous-subtype PDA tumors (*Bailey et al., 2016*). We next interrogated single-cell RNA-seq data from 24 human PDA tumors and 11 normal pancreas tissue samples from a recently published study (*Peng et al., 2019*). This analysis confirmed that *TP63* is aberrantly expressed in the ductal compartment of a subset of patient samples that exhibit high-level expression of squamous marker genes (*Figure 6B–C* and *Figure 6—figure supplement 1A–B*). Importantly, we observed significantly higher expression of inflammatory genes (*CXCL1*, *IL6* and *LIF*) within the fibroblast compartment in *TP63*high versus *TP63*low patients (*Figure 6D* and *Figure 6—figure supplement 1C*). Finally, we performed a multiplex immunofluorescence staining of 12 human PDA tissue samples, which revealed a trend towards increased neutrophil infiltration in p63-positive versus p63-negative PDA samples (*Figure 6E–F* and *Figure 6—figure supplement 1D*).

## Discussion

Prior studies have revealed PDA to be a heterogeneous disease at the molecular level with respect to both the tumor and stromal compartments (*Bailey et al., 2016*; *Cancer Genome Atlas Research Network, 2017*; *Collisson et al., 2011*; *Moffitt et al., 2015*). As both compartments are comprised of cell types that display phenotypic plasticity (*Öhlund et al., 2017*; *Yuan et al., 2019*), dissecting the molecular cross-talk between cells that define each patient's tumor remains a formidable challenge. Although prior studies have made correlations between epithelial and stromal transcriptomes

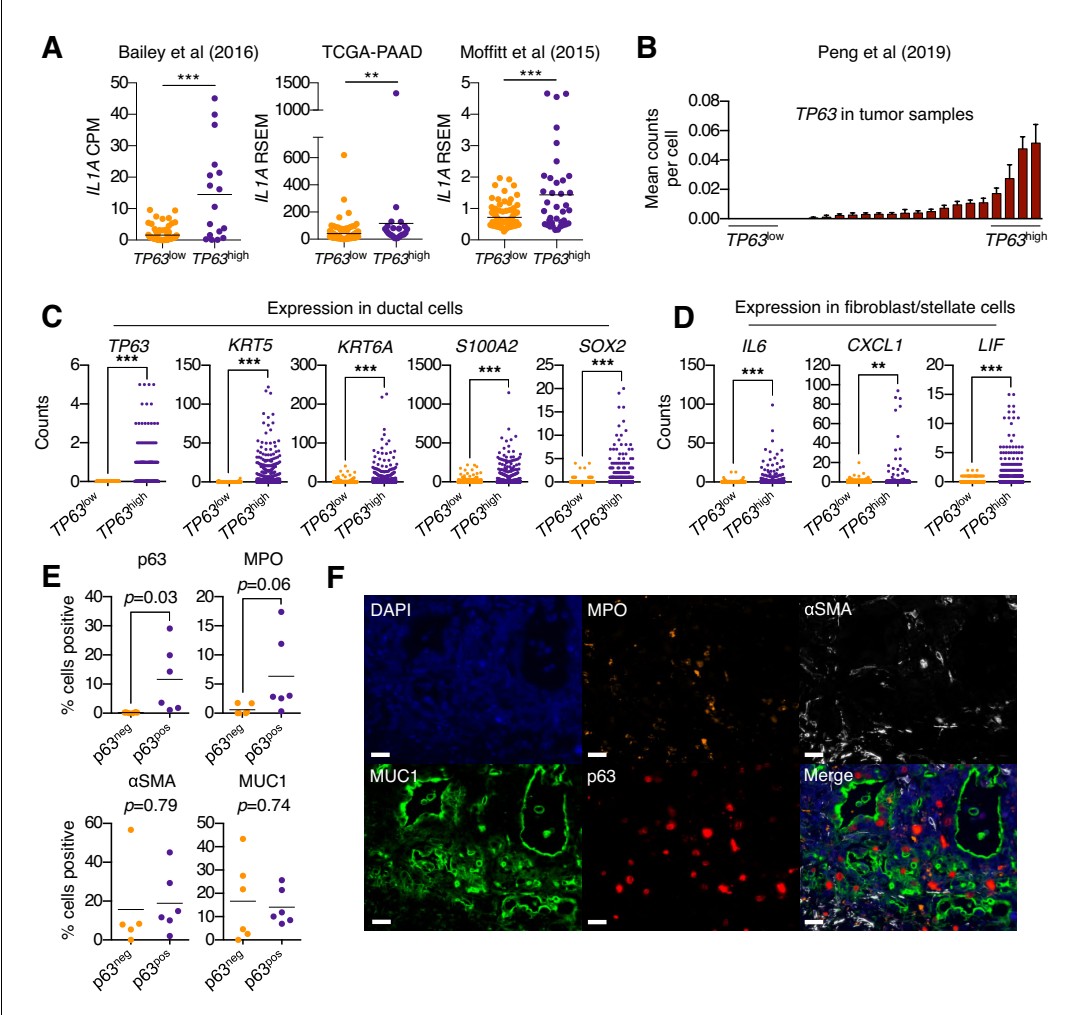

**Figure 6.** Squamous trans-differentiation of PDA cells correlates with increased inflammatory fibroblasts and neutrophil infiltration in patient samples. (A) Quantification of *IL1A* mRNA levels in the indicated studies. Each dot represents one patient sample, and patients were stratified as *TP63*high or *TP63*low as described in *Somerville et al. (2018)*. (B–D) Single cell RNA-seq data from 24 primary PDA tumor and 11 normal pancreas samples from the study by *Peng et al. (2019)*. (B) 24 primary PDA tumor samples ranked by mean *TP63* expression levels across all cell types. Patients were stratified as *TP63*high (n = 4) or *TP63*low (n = 4) as shown. (C) Expression of squamous genes in ductal cells from *TP63*high or *TP63*low PDA patient samples from (B). (D) Expression of inflammatory genes (*IL6*, *CXCL1* and *LIF*) in stellate cells and fibroblast cells from *TP63*high or *TP63*low PDA patient samples from (B). Each dot represents a single cell and the four patient samples were pooled for this analysis. (E–F) Multiplex immunofluorescent staining of primary pancreatic cancer tumor from the Rapid Autopsy Program. (E) Quantification of p63, MPO, αSMA and MUC1 in p63-negative (p63neg, n = 6) and p63-positive (p63pos, n = 6) samples. (F) Representative images of a p63-positive PDA tumor stained with DAPI (blue), MPO (orange), αSMA (white), MUC1 (green), p63 (red), and overlaid (merge). The images were collected from the same tumor area, but over multiple rounds of iterative staining. Scale bar indicates 20 μm. For (A), (C) and (D) ***p<0.001, **p<0.01 by Student's t-test. ns, not significant.

The online version of this article includes the following figure supplement(s) for figure 6:

**Figure supplement 1.** p63 expression in PDA cells is associated with inflammatory CAF induction and neutrophil infiltration in patient samples.

in PDA (*Maurer et al., 2019*; *Moffitt et al., 2015*; *Puleo et al., 2018*), it remains unclear whether the evolution of these two compartments is dependent on one another.

Here, we have defined a mechanism by which tumor cells gain an ability to reprogram their local microenvironment through the acquisition of p63, which is a master regulator of the squamous sub-type of PDA (*Hamdan and Johnsen, 2018*; *Somerville et al., 2018*). Mechanistically, we have shown that p63 activates a pro-inflammatory secretory phenotype that triggers iCAF induction through an IL-1-dependent signaling mechanism. Importantly, we have verified the correlation between *TP63* and *IL1A* expression in three independent transcriptome studies of human PDA tumors. In addition,

in independent studies of primary PDA patient samples, we have shown that squamous trans-differentiation of human pancreatic cancer cells is associated with an enhanced inflammatory CAF phenotype and increased neutrophil infiltration. A recent transcriptome study of human tissue samples also revealed a correlation between squamous-subtype PDA tumors and inflammation-associated changes in the stroma (*de Santiago et al., 2019*), which validates that our observations in experimental models are in accord with clinical observations. While the aberrant expression of p63 confers a myriad of tumor cell-intrinsic phenotypes in PDA, such as enhanced motility, invasion, and resistance to cytotoxic chemotherapy (*Danilov et al., 2011*; *Somerville et al., 2018*), our work points to a powerful non cell-autonomous effect of p63 in PDA in promoting stromal inflammation.

We previously demonstrated that IL-1/JAK/STAT signaling is the key pathway responsible for iCAF formation in PDA (*Biffi et al., 2019*). While a basal level of IL-1 is expressed in PDA tumor cells that lack p63 expression (*Biffi et al., 2019*; *Tjomsland et al., 2011*; *Zhang et al., 2018*), a key finding in our current study is in demonstrating that a striking heterogeneity exists in IL-1 expression in the human disease, with the highest levels being found in p63-expressing squamous-subtype tumors. We account for this observation by demonstrating that a super-enhancer is installed by p63 at the *IL1* locus, which enables a > 30 fold increase in IL-1α secretion to enhance iCAF formation. One expected outcome of p63-mediated *IL1A* hyperactivation would be a strengthened concentration gradient of IL-1α emanating from tumor cells to promote iCAF induction within the stroma. Whilst this hypothesis is robustly supported in the current study, our in vitro and in vivo studies, together with single cell RNA-seq analysis of PDA patient tumors, suggest that with increasing complexity of the tumor microenvironment, the association of p63 expression in PDA cells with myCAF abundance is more variable. While the abundance and spatial localization of iCAFs and myCAFs are likely regulated by several signaling pathways (e.g. TGF-β), our in vivo findings suggest that the acquisition of p63 expression in the tumor leads to a potent increase in IL-1 secretion to drive iCAF enrichment in the surrounding microenvironment. Interestingly, a recent study identified a critical in vivo role for tumor cell-derived IL-1β in inflammatory CAF activation and immune suppression using murine orthotopic models of PDA (*Das et al., 2020*). Mechanistically, the expression of IL-1β in vivo was shown to be driven by toll-like receptor 4 (TLR4) activation and subsequent engagement of the NLRP3 inflammasome. Taken together with our findings, these data suggest that the transcriptional activation of both *IL1A* and *IL1B* may be functionally relevant within the tumor microenvironment of squamous-subtype PDA tumors.

Many of the genes regulated by p63 in PDA cells correspond to those activated by the pro-inflammatory transcription factor NF-κB. Indeed, it has been previously shown that p63 and NF-κB co-occupy a common set of target genes in head and neck squamous cell carcinoma, and this is associated with enhanced inflammation in vivo (*Yang et al., 2011*). In addition, NF-κB is often constitutively active in most pancreatic cancer tissues and confers resistance to apoptosis (*Dong et al., 2002*; *Wang et al., 1999*). In this context, constitutive activation of NF-κB requires an IL-1α-driven feed-forward amplification loop to promote PDA development (*Ling et al., 2012*; *Niu et al., 2004*). Maintenance of an inflamed CAF state has also been shown to be dependent on the NF-κB signaling pathway (*Biffi et al., 2019*; *Erez et al., 2010*). Taken together, these studies raise the possibility that p63 functions as an amplifier of NF-κB transcriptional output to drive enhanced secretion of inflammatory mediators in squamous-subtype PDA tumors. In addition to IL-1α, p63 promotes expression of other pro-inflammatory cytokines that have been implicated in the development and progression of PDA, such as CXCL1 and GM-CSF (*Bayne et al., 2012*; *Li et al., 2018*; *Pylayeva-Gupta et al., 2012*). Notably, both of these cytokines are also produced by iCAFs (*Öhlund et al., 2017*), suggesting that the pro-tumorigenic effects of these factors would be amplified by IL-1α-mediated crosstalk between p63-expressing tumor cells and iCAFs.

An association between squamous cell carcinomas and infiltrating inflammatory cells has been observed previously in other tumor contexts (*Andreu et al., 2010*; *Coussens et al., 1999*; *Erez et al., 2010*; *Eruslanov et al., 2014*; *Ferone et al., 2016*; *Kargl et al., 2017*; *Xu et al., 2014*; *Yang et al., 2011*). In one study, squamous trans-differentiation in a lung adenocarcinoma mouse model was associated with neutrophil infiltration and other markers of inflammation (*Mollaoglu et al., 2018*). In this context, inflammatory changes were promoted by the squamous lineage transcription factor SOX2 through activation of CXCL5 expression (*Mollaoglu et al., 2018*). It is interesting to note that we also observed a correlation between *TP63* and *SOX2* expression in single cell RNA-seq analysis of tumor ductal cells from PDA patent samples. Taken together with our

findings, these results suggest that inflammation is more generally associated with squamous lineage tumors through multiple distinct mechanisms. Interestingly, a major function of the normal squamous epithelium is to serve as a protective barrier to exogenous insults. As such, inflammatory pathways must be tightly regulated and poised for a rapid response in these tissues. Notably, squamous cells of the epidermis are known to express large amounts of IL-1α at steady state (*Hauser et al., 1986*) and *IL1A* is known to be a key p63 target in normal human keratinocytes (*Barton et al., 2010*). For this reason, amplified inflammation may be an inevitable consequence of tumor trans-differentiation into the squamous lineage in diverse tissue contexts.

Histopathological definitions of PDA are often imprecise and offer limited clinical utility with respect to patient management and treatment selection (*Klöppel and Lüttges, 2001*). Adenosquamous carcinoma, for example, is defined by those tumors exhibiting >15% squamous differentiation and is considered an uncommon variant of PDA. Studies that have identified molecular subtypes of PDA using transcriptomic analyses provide the opportunity to define a new molecular taxonomy for pancreatic cancer that is more clinically applicable (*Collisson et al., 2019*). The challenge now is to understand the underlying biology that renders these molecular subtypes phenotypically distinct and to identify robust markers for prospective patient stratification. The findings presented in our study point to p63 as a functional biomarker that identifies a poor prognostic subgroup of PDA patients in which inflammatory pathways are amplified. As such, these patients may benefit from targeted therapies that aim to dampen the inflammatory response. For example, monoclonal antibodies targeting IL-1α or IL-1β have shown promise as potential cancer therapies (*Hickish et al., 2017*; *Hong et al., 2014*; *Ridker et al., 2017*), and the IL-1 receptor antagonist Anakinra has shown efficacy in pre-clinical models of PDA (*Zhuang et al., 2016*). The findings presented in this study suggest that such agents may also show efficacy in appropriately stratified PDA patients.

# Materials and methods

**Key resources table**

| Reagent type (species) or resource | Designation | Source or reference | Identifiers | Additional information |
|---|---|---|---|---|
| Genetic reagent (*Mus. musculus*) | NOD.Cg-*Prkdc*scid *Il2rg*tm1Wjl/SzJ (NSG) | The Jackson Laboratory | Stock #: 005557, RRID:IMSR_JAX:005557 | |
| Genetic reagent (*Mus. musculus*) | C57BL/6J | The Jackson Laboratory | Stock #: 000664, RRID:IMSR_JAX:000664 | |
| Cell line (*Homo sapiens*) | PANC1 | ATCC | Cat# CRL-1469, RRID:CVCL_0480 | |
| Cell line (*Homo sapiens*) | PATU8988S | DSMZ | Cat# ACC-204, RRID:CVCL_1846 | |
| Cell line (*Homo sapiens*) | SUIT2 | JCRB | JCRB1094, RRID:CVCL_3172 | |
| Cell line (*Homo sapiens*) | CFPAC1 | ATCC | Cat# CRL-1918, RRID:CVCL_1119 | |
| Cell line (*Homo sapiens*) | AsPC1 | ATCC | CRL-1682, RRID:CVCL_0152 | |
| Cell line (*Homo sapiens*) | PANC0403 | ATCC | Cat# CRL-2555, RRID:CVCL_1636 | |
| Cell line (*Homo sapiens*) | CAPAN2 | ATCC | Cat# HTB-80, RRID:CVCL_0026 | |
| Cell line (*Homo sapiens*) | HPAFII | ATCC | Cat# CRL-1997, RRID:CVCL_0313 | |
| Cell line (*Homo sapiens*) | MIAPaca2 | ATCC | Cat# CRL-1420, RRID:CVCL_0428 | |
| Cell line (*Homo sapiens*) | BxPC3 | ATCC | Cat# CRL-1687, RRID:CVCL_0186 | |

*Continued on next page*

*Continued*

| Reagent type (species) or resource | Designation | Source or reference | Identifiers | Additional information |
|---|---|---|---|---|
| Cell line (*Homo sapiens*) | T3M4 | RCB | Cat# RCB1021, RRID:CVCL_4056 | |
| Cell line (*Homo sapiens*) | KLM1 | RCB | Cat# RCB2138, RRID:CVCL_5146 | |
| Cell line (*Homo sapiens*) | Human PSCs | PMID:30366930 | | Primary |
| Cell line *Mus musculus* | mPSC4; mPSC5 | PMID:30366930 | | Primary |
| Cell line *Mus musculus* | mM1 | PMID:25557080 | | Primary |
| Antibody | anti-p63 (rabbit monoclonal) | Cell Signaling | Cat# 39692, RRID:AB_2799159 | 1:1000 (WB) 1:500 (IHC) 1:500 (IF) |
| Antibody | anti-HSC70 (mouse monoclonal) | Santa Cruz Biotechnology | Cat# sc-7298, RRID:AB_627761 | 1:5000 (WB) |
| Antibody | anti-ACTB (mouse monoclonal) | Sigma-Aldrich | Cat# A3854, RRID:AB_262011 | 1:1000 (WB) |
| Antibody | anti-Cas9 (mouse monoclonal) | Epigentek | Cat# A-9000–050, RRID:AB_2828022 | 1:1000 (WB) |
| Antibody | anti-MPO (goat polyclonal) | R and D Systems | Cat# AF3667, RRID:AB_2250866 | 1:200 (IF) |
| Antibody | anti-LY6G (rat monoclonal) | BD Pharmingen | Cat# 551459, RRID:AB_394206 | 1:200 (IF) |
| Antibody | anti-MPO (rabbit polyclonal) | Abcam | Cat# ab9535, RRID:AB_307322 | 1:200 (IF) |
| Antibody | anti-alpha-SMA (mouse monoclonal) | Sigma-Aldrich | Cat# A2547, RRID:AB_476701 | 1:200 (IF) |
| Antibody | anti-Muc1 (Armenian hamster monoclonal) | Abcam | Cat# ab80952, RRID:AB_1640314 | 1:200 (IF) |
| Antibody | anti-CD45-PerCP-Cy5-5 (rat monoclonal) | BioLegend | Cat# 103132, RRID:AB_893340 | 1:100 (FC) |
| Antibody | anti-PDPN-APC/Cy7 (Syrian hamster monoclonal) | BioLegend | Cat# 127418, RRID:AB_2629804 | 1:100 (FC) |
| Antibody | anti-CD326 (EPCAM)-AlexaFluor 647 (mouse monoclonal) | BioLegend | Cat# 324212, RRID:AB_756086 | 1:100 (FC) |
| Antibody | anti-E-Cadherin-AlexaFluor 647 (rat monoclonal) | BioLegend | Cat# 147304, RRID:AB_2563040 | 1:100 (FC) |
| Antibody | anti-CD45-BV510 (rat monoclonal) | BioLegend | Cat# 103138, RRID:AB_2563061 | 1:100 (FC) |
| Antibody | anti-F4/80-BV785 (rat monoclonal) | BioLegend | Cat# 123141, RRID:AB_2563667 | 1:100 (FC) |
| Antibody | anti-Ly6G/Ly6C (Gr-1)-PE (rat monoclonal) | BioLegend | Cat# 108408, RRID:AB_313373 | 1:100 (FC) |
| Antibody | anti-CD11b-PE-Cy7 (rat monoclonal) | BioLegend | Cat# 101216, RRID:AB_312799 | 1:500 (FC) |
| Antibody | anti- CD8a-APC-Cy7 (rat monoclonal) | BioLegend | Cat# 100714, RRID:AB_312753 | 1:100 (FC) |
| Antibody | anti-CD4-APC (rat monoclonal) | BioLegend | Cat# 100516, RRID:AB_312719 | 1:100 (FC) |

*Continued on next page*

*Continued*

| Reagent type (species) or resource | Designation | Source or reference | Identifiers | Additional information |
|---|---|---|---|---|
| Antibody | anti-TCRβ-PE-Cy7 (Armenian hamster monoclonal) | BioLegend | Cat# 109222, RRID:AB_893625 | 1:500 (FC) |
| Antibody | Anti-CD11b-AlexaFluor 488 (rat monoclonal) | BioLegend | Cat# 101217, RRID:AB_389305 | 1:100 (FC) |
| Antibody | anti-CXCL1 (mouse monoclonal) | R and D Systems | Cat# MAB275, RRID:AB_2292460 | 10 µg/ml |
| Antibody | IgG2B Isotype Control (mouse monoclonal) | R and D Systems | Cat# MAB004, RRID:AB_357346 | 10 µg/ml |
| Recombinant DNA reagent | LentiV-Cas9-puro | Addgene | Cat# 108100 | |
| Recombinant DNA reagent | LRNG vector | Addgene | Cat# 125593 | |
| Recombinant DNA reagent | Lenti-dCas9-KRAB-blast | Addgene | Cat# 89567 | |
| Recombinant DNA reagent | YXP-Cas9-puro | This paper | | See 'CRISPR-based Targeting' in Materials and methods |
| Recombinant DNA reagent | Lenti-luciferase-blast vector | PMID:30428345 | | |
| Recombinant DNA reagent | LentiV-ΔNp63-neo vector | PMID:30428345 | | |
| Peptide, recombinant protein | mouse CXCL1 | Peprotech | Cat# 250–11 | |
| Peptide, recombinant protein | human CXCL1 | Peprotech | Cat# 300–11 | |
| Commercial assay or kit | IL-1α ELISA assay | R and D Systems | Cat# DLA50 | |
| Commercial assay or kit | IL-1β ELISA assay | R and D Systems | Cat# DLB50 | |
| Commercial assay or kit | IL-6 ELISA assay | R and D Systems | Cat# M6000B | |
| Other | Matrigel | Corning | Cat# 356231 | |
| Other | 3 µm FluoroBlok cell culture insert | Corning | Cat# 08-772-141 | |
| Software, algorithm | HISAT2 | PMID:25751142 | HISAT2, RRID:SCR_015530 | |
| Software, algorithm | Cuffdiff | PMID:20436464 | Cuffdiff, RRID:SCR_001647 | |
| Software, algorithm | Morpheus | Broad Institute | Morpheus by Broad Institute, RRID:SCR_017386 | |
| Software, algorithm | Metascape | PMID:30944313 | Metascape, RRID:SCR_016620 | |
| Software, algorithm | GSEA | PMID:16199517 | Gene Set Enrichment Analysis, RRID:SCR_003199 | |
| Software, algorithm | DESeq2 | PMID:25516281 | DESeq2, RRID:SCR_015687 | |
| Software, algorithm | SAMtools 1.4 | PMID:19505943 | SAMTOOLS, RRID:SCR_002105 | |

## Cell lines and cell culture

Mouse PSCs and pancreatic organoid lines were previously described (*Boj et al., 2015*; *Öhlund et al., 2017*). Human PSCs were purchased from ScienCell (3830). Mouse PSCs in which the

*Il1r1* was knocked out using CRISPR were previously described (*Biffi et al., 2019*). Mouse and human PSCs were cultured in DMEM (10–013-CV; Fisher Scientific) containing 5% FBS. HEK 293 T cells were cultured in DMEM containing 10% FBS. Human PDA cell lines were cultured in RPMI 1640 (Gibco) containing 10% FBS. All cells were cultured with 1% L-glutamine and 1% Penicillin/Streptomycin at 37°C with 5% $CO_2$. All cell lines were routinely tested for mycoplasma and were authenticated using STR profiling. For conditioned media experiments, human PDA cell lines or mouse organoids were cultured for 3–4 days in DMEM containing 5% FBS before media were collected and filtered through a 0.45 µm syringe filter to remove debris. Conditioned media were stored at 4°C for no more than 3–4 days or snap frozen and stored at −80°C prior to use.

## Lentiviral production and infection

Lentivirus was produced in HEK 293 T cells by transfecting plasmid DNA and packaging plasmids (VSVG and psPAX2) using Polyethylenimine (PEI 25000; Polysciences; Cat# 23966–1). Media were replaced with target media 6–8 hr following transfection and lentivirus-containing supernatant was subsequently collected every 12 hr for 48 hr prior to filtration through a 0.45 µm filter. For infection of human PDA cells, cell suspensions were mixed with lentiviral-containing supernatant supplemented with polybrene to a final concentration of 4 µg/ml. Cells were plated in tissue culture plates of the appropriate size and lentiviral-containing supernatant was replaced with fresh media after an incubation period of 24 hr. For infection of pancreatic organoids, lentivirus was first concentrated 10x in mouse organoid media (*Boj et al., 2015*) using Lenti-X concentrator according to the manufacturer's instructions. Organoid cultures were then dissociated into single cells, and spinoculated by centrifugation as described previously (*Roe et al., 2017*).

## CRISPR-based targeting

To generate cell lines in which *TP63* had been stably knocked out, KLM1 cells expressing Cas9 in the LentiV-Cas9-puro vector (addgene # 108100) were infected in a pooled fashion with control or p63 sgRNAs in the LRNG vector (*Roe et al., 2017*). Two days post infection with sgRNAs, transduced cells were selected with 1 mg/ml of G418 for five days before they were used for conditioned media experiments or orthotopic transplantations. For GFP-depletion assays, KLM1 and T3M4 cells were infected with sgRNAs as described above but without selection and GFP% was measured on day three (P0) and then every three days post-viral transduction until day 18 (P5). For CRISPR interference at the IL1 enhancer, SUIT2-empty or SUIT2-p63 cells stably expressing catalytically-dead Cas9 fused with a KRAB repression domain in the Lenti-dCas9-KRAB-blast vector (addgene #89567) were infected in a pooled fashion with sgRNAs targeting IL1 enhancer regions (identified from p63 and H3K27ac ChIP-seq in these lines) in the LRNG vector. Stable cell lines were selected by G418 selection for seven days prior to their use in conditioned media experiments. For doxycycline-inducible CRISPR based targeting of p63 in T3M4 and KLM1 cells, Cas9 from the LentiV-Cas9-puro vector was cloned into the doxycycline-regulated TREtight-cDNA-EFS-rtTA-P2A-Puro vector (*Somerville et al., 2018*) to generate the YXP-Cas9-puro vector. T3M4 and KLM1 cells were infected with the YXP-Cas9-puro vector, selected with 3 µg/ml puromycin to generate stable cell line before infection with sgRNAs targeting *TP63* (sg*TP63*#4) or a control sgRNA (sgNEG) in LRNG vector. Two days post infection with sgRNAs, transduced cells were selected with 1 mg/ml of G418 for three days and were subsequently treated with doxycycline (1 µg/ml) for 48 hr prior to harvesting RNA and lysate for RNA-seq and western blot analysis, respectively. sgRNA sequences can be found in *Supplementary file 5*.

## RNA extraction and RT-qPCR analysis

Total RNA was extracted using TRIzol reagent following the manufacturer's instructions. For RNA extraction from cells cultured in Matrigel, cells were lysed by adding TRIzol reagent directly to the Matrigel dome. 200 ng-1µg of total RNA was reverse transcribed using qScript cDNA SuperMix (Quanta bio; 95048–500), followed by RT-qPCR analysis with Power SYBR Green Master Mix (Thermo Fisher Scientific; 4368577) on an ABI 7900HT fast real-time PCR system. Gene expression was normalized to *GAPDH* or *Gapdh* for human and murine gene expression analysis, respectively. RT-qPCR primers used can be found in *Supplementary file 5*.

## Cell lysate preparation and western blot analysis

Cell cultures were collected and 1 million cells were counted by trypan blue exclusion and washed with ice cold PBS. Cells were then resuspended in 100 µl PBS and lysed with 100 µl of 2x Laemmli Sample Buffer supplemented with β-mercaptoethanol by boiling for 30 min. Samples were centrifuged at 4°C for 15 min at 10,000xg and the supernatant was used for western blot analysis with standard SDS-PAGE-based procedures. Primary antibodies used were p63 (Cell Signaling; 39692), HSC70 (Santa Cruz; sc-7298), ACTB (Sigma-Aldrich; A3854) and Cas9 (Epigentek; A-9000–050) and proteins were detected using HRP-conjugated secondary antibodies.

## Elisa

For ELISA of plasma, blood was harvested via cardiac puncture from anesthetized mice at end point and plasma separated by centrifugation. For ELISA of media, conditioned media was harvested as described above. ELISA assays used were IL-1α (DLA50; R and D Systems), IL-1β (DLB50; R and D Systems) and IL-6 (M6000B; R and D Systems).

## In Vitro luciferase imaging

To generate luciferase expressing mouse PSC cultures, cells were infected with a luciferase transgene in a Lenti-luciferase-blast vector (*Somerville et al., 2018*) and stable cell lines were generated by selection with 10 µg/ml blasticidin. Cells were plated at a density of 10,000 cells per 20 µl of Matrigel in each well of a black, clear-bottom, ultra-low attachment 96-well plate (10014–318; CELL-STAR) and 200 µl of conditioned media was added. Cells were imaged following four days of culture using an IVIS Spectrum system (Caliper Life Sciences) six minutes post addition of D-Luciferin (150 µg/ml) to each well. For luciferase-based proliferation assays of SUIT2-p63 cells in vitro, parental SUIT2 cells were first infected with Lenti-luciferase-blast vector and stable SUIT2-luciferase cell lines were generated by selection with 10 µg/ml blasticidin. SUIT2-luciferase cells were then infected with p63 cDNA in LentiV-ΔNp63-neo vector (*Somerville et al., 2018*) or the empty vector as a control. Two days post infection, transduced cells were selected with 1 mg/ml of G418 and on day seven post infection, cells were counted by trypan blue exclusion and seeded at a density of 500 cells per well in 200 µl of media in black 96-well plates (137101; Thermo). Cells were imaged daily as described above.

## In Vivo orthotopic transplantations and bioluminescence imaging

Transplantation of human PDA cells and mouse pancreatic organoid cultures have been described previously (*Boj et al., 2015*; *Somerville et al., 2018*). For transplantation of human cells, NSG mice were used (NOD.Cg-*Prkdc*scid *Il2rg*tm1Wjl/SzJ). For transplantation of mouse cells, C57BL/6J mice were used. For mM1 organoid transplantations, $1 \times 10^5$ cells were prepared from organoid cultures as a 45 µL suspension of 50% Matrigel in PBS and injected into the pancreas. For p63 knockout experiments in vivo, KLM1-Cas9 cells were first infected with Lenti-luciferase-blast vector and a stable KLM1-Cas9-luciferase cell line was generated by selection with 10 µg/ml blasticidin. These cells were subsequently infected with control or p63 sgRNAs in LRNG vectors as described above. G418-selected cells were counted by trypan blue exclusion and $5 \times 10^4$ viable cells were prepared as a 50 µL suspension of 50% Matrigel in PBS and injected into the pancreas. For in vivo experiments using SUIT2-luciferase cells, these cells were infected with LentiV-ΔNp63-neo vector or the empty vector as a control and two days post infection, transduced cells were selected 1 mg/ml of G418 for five days and on day seven post infection, G418-selected cells were counted by trypan blue exclusion and $2 \times 10^4$ viable cells were prepared as a 50 µL suspension of 50% Matrigel in PBS and injected into the pancreas. For bioluminescence imaging, mice were intraperitoneally (IP) injected with D-Luciferin (50 mg/kg) and analyzed using an IVIS Spectrum system (Caliper Life Sciences) ten minutes post IP injection.

## Histology and immunohistochemistry

Histological analysis and p63 immunohistochemistry (IHC) was performed as previously described (*Somerville et al., 2018*). Primary antibodies used were p63 (Cell Signaling; 39692). Hematoxylin and eosin and Masson's trichrome staining were performed according to standard protocols. Stained sections were scanned with Aperio ScanScope CS and to quantify p63, the percentage of strong

positive nuclei was calculated relative to the total number of nuclei with the ImageScope nuclear v9 algorithm.

## Immunofluorescence staining

To stain for myeloperoxidase and Ly6G, paraffin sections were deparaffinized and rehydrated, and antigen retrieval was carried out by boiling slides in Tris EDTA buffer (10 mM Tris Base, 1 mM EDTA, pH 9.0) for 6 min in a pressure cooker. The slides were blocked with 1x blocking buffer (PBS containing 2.5% BSA and 5% goat serum) and Fc receptor blocker (Innovex Biosciences, Richmond, CA), before incubating with anti-myeloperoxidase (1:200, AF3667, R and D) or anti-Ly6G (1:200, #551459, BD Pharmingen) antibody overnight at 4℃. After two washes in PBS, slides were incubated in the presence of fluorochrome-conjugated secondary antibodies (1:250, Invitrogen) for 1 hr, and counterstained with DAPI (1:200 dilution; Life Technologies). Images were collected at 40x magnification using a Leica SP8 confocal and were analyzed using ImageJ software.

## Multiplex immunofluorescence staining of human PDA patient tissues

Human tissue samples were collected from end stage PDA patients within 3 hr of the patient passing at the Rapid Autopsy Pancreatic Program (RAPP) at the University of Nebraska Medical Center. At autopsy, pathologists identified primary tumor and metastatic sites grossly and samples were collected from tumor, adjacent, and uninvolved tissue. Samples were stored in formalin for rapid paraffin imbedding. 5µm-thick sections were cut and mounted onto charged slides. Immunofluorescent analysis was then performed as follows. The antibody recognizing MPO (Abcam: ab9535) was directly conjugated to fluorescent moieties using Cy5 labeling kits obtained from GE Health (Amersham CyDye kits). Alpha-SMA was purchased directly conjugated to Cy3 (Sigma Aldrich: A2547). Secondary antibodies were used to identify p63 (Cell Signaling: 39692) and Muc1 (Abcam: ab80952). The slides were then deparaffinized and rehydrated using xylene and an alcohol gradient. Antigen retrieval was performed using heated acidic citrate buffer for twenty minutes at low heat in a microwave and the slides were blocked in 1% BSA. The slides were stained with directly labeled antibodies or primary and secondary antibodies as needed. Coverslips were mounted using mounting media with DAPI and allowed to dry for 24 hr before scanning. Following slide scanning, the slides were decoverslipped via inversion in agitated PBS. Once the cover slip was removed the slides were submerged in 3% Triton for ten minutes. Slides were then left in an agitated alkaline $H_2O_2$ solution for fifteen minutes to quench the fluorescent molecule and allow for restaining with a new set of directly labeled antibodies. Three rounds of staining were performed. Slide scanning was performed using a Pannoramic 250 whole slide scanner at x20 (3D Histech). Slides were scanned with the test tissue and a sequential control cut, using consistent exposure for each antibody. Once slides were scanned, they were analyzed using HALO (Indica Labs), a whole slide image analysis software, to determine the exposure of each marker across the tumor microenvironment. Calling a positive 'hit' was based on the average for visual thresholding across five different samples and then applied to all tissue with that antibody stain. These data were used to determine the percentage of cells expressing each marker. A random forest masking algorithm was used to remove autofluorescence associated with tissue damage. Only the tumor, and not the surrounding adjacent normal or necrotic tissue, was analyzed.

## Flow cytometry and cell sorting

Tumors were processed as previously described (Öhlund et al., 2017). For sorting of cancer cells, fibroblasts and immune cells from xenografted tumors, cells were stained for 30 min with anti-mouse CD45-PerCP-Cy5-5 (103132; BioLegend), PDPN-APC/Cy7 (127418; BioLegend) and anti-human CD326 (EPCAM)-AlexaFluor 647 (324212; BioLegend), anti-human/mouse E-Cadherin- AlexaFluor 647 (147304; BioLegend) and DAPI for 15 min. Cells were sorted on the FACSAria cell sorter (BD) for DAPI/CD45⁻EPCAM/E-Cadherin⁺ (cancer cells), DAPI/CD45/EPCAM/E-Cadherin⁻PDPN⁺ (fibroblasts) and DAPI⁻CD45⁺ (immune cells) cell populations. Sorted cells were pelleted and resuspended in TRIzol for RNA extraction. For flow cytometric analysis of processed mM1 tumors, antibodies used for analysis of myeloid cells were: anti-mouse CD45-BV510 (103138; BioLegend), F4/80-BV785 (123141; BioLegend), Ly6G/Ly6C (Gr-1)-PE (108408; BioLegend) and CD11b-PE-Cy7 (101216; BioLegend); antibodies used for analysis of T-cells cells were: anti-mouse CD45-BV510 (103138;

BioLegend), CD8a-APC-Cy7 (100714; BioLegend), CD4-APC (100516; BioLegend) and TCRβ-PE-Cy7 (109222; BioLegend). For flow cytometric analysis of processed xenografted tumors, antibodies used were anti-mouse CD45-PerCP-Cy5.5 (103132; BioLegend), F4/80-BV785 (123141; BioLegend), Ly6G-Ly6C (Gr-1)-PE (108408; BioLegend) and CD11b-AlexaFluor 488 (101217; BioLegend). Fixable viability dye eFluor450 (eBioscience) was used to differentiate between live and dead cells.

## Neutrophil isolation and transwell migration assays

Neutrophil isolation was performed as previously described (*Albrengues et al., 2018*). Transwell migration assays were performed with conditioned media derived from cell lines ectopically expressing p63 or the empty vector that were cultured alone or in combination with mouse PSCs for five days in Matrigel. $5 \times 10^3$ SUIT2 cells and $4 \times 10^4$ PSCs were used for the co-cultures. The conditioned media were added to the lower chamber, and primary mouse neutrophils ($2.5 \times 10^5$) were seeded in 0.5% FBS containing DMEM in the upper chamber of a 3 μm FluoroBlok cell culture insert (08-772-141; Corning) in a 24 well plate. 500 ng of recombinant mouse (250-11; Peprotech) or human (300-11; Peprotech) CXCL1 was added to DMEM with 5% FBS on the day of seeding neutrophils as controls. For experiments with blocking antibodies, 10 μg/ml human CXCL1 (MAB275-100; R and D Systems) or IgG2B isotype control (MAB004; R and D Systems) was added to the neutrophils prior to the addition of the conditioned media. After 24 hr, the FluoroBlok membrane was stained with DAPI (0.05 mg/ml; D1306; Thermo Fisher Scientific) for 5 min, rinsed in water and mounted onto glass slides using mounting media (17985–16; Electron Microscopy Sciences). The number of invading neutrophils was counted in 10 random fields of view using a fluorescence microscope (Leica SP8 Confocal microscope).

## RNA-seq library construction

RNA-seq libraries were constructed using the TruSeq sample Prep Kit V2 (Illumina) according to the manufacturer's instructions. Briefly, 0.5–2 μg of purified RNA was poly-A selected and fragmented with fragmentation enzyme. cDNA was synthesized with Super Script II Reverse Transcriptase (Thermo Fisher; 18064014), followed by end repair, A-tailing and PCR amplification. RNA-seq libraries were single-end sequenced for 50 bp, or for xenograft RNA-seq analysis paired-end sequenced for 150 bp, using an Illumina NextSeq platform (Cold Spring Harbor Genome Center, Woodbury).

## RNA-Seq data analysis

Single end 50 bp sequencing reads were mapped to the mm10 or hg38 genomes using HISAT2 with standard parameters (*Kim et al., 2015*). Structural RNA was masked and, unless stated otherwise, differentially expressed genes were identified using Cuffdiff (*Trapnell et al., 2010*). All the following analysis was performed on genes with an RPKM value no less than two in either control or experimental samples. Heat maps of standardized expression values for differentially expressed genes were generated using Morpheus from the Broad Institute (https://software.broadinstitute.org/morpheus). To generate ranked gene lists for Pre-ranked GSEA, genes were ranked by their mean $\log_2$ fold change between the two experimental groups of interest. Gene ontology analysis was performed using Metascape (*Zhou et al., 2019*). For clustering of mouse PSC cultures, the $\log_2$(RPKM +1) values for each cell line was used to generate a heat map of similarity matrix by Pearson correlation, which was subsequently clustered by Euclidean distance with average linkage using Morpheus. Group 1 and Group 2 cultures were then analyzed using Cuffdiff and those genes with $\log_2$ fold change >1 and a FDR value <0.05 were considered and differentially expressed. For GSEA using the Broad Institutes Hallmark gene sets, mouse gene symbols were converted to their orthologous human symbols and ranked by their average $\log_2$ fold change in the respective cultures. For RNA-seq analysis of flow sorted cancer cells, fibroblasts and immune cells, those genes with a FDR value <0.05 when comparing sg*TP63*#4 versus sgNEG tumors were considered and differentially expressed. For xenograft RNA-seq analysis, paired end 150 bp sequencing reads were mapped to the mm10 or hg38 genomes using HISAT2 with standard parameters. The alignment sam files were converted to bam format using the samtools view command. To disambiguate between the human and mouse reads, Disambiguate (*Ahdesmäki et al., 2016*) was run on the bam files, and the unambiguous reads were used for RNA quantification and differential expression analysis using DESeq2 (*Love et al., 2014*). For human transcripts, those genes with $\log_2$ fold change >1 and a FDR

value <0.1 were considered as differentially expressed. For mouse transcripts, those genes with a FDR value <0.1 were considered as differentially expressed. To generate a ranked list of mouse genes for GSEA analysis genes the bottom 30% of expressed genes across all groups were removed. The iCAF and myCAF gene signatures were defined as the top 200 up- and down-regulated genes in iCAFs versus myCAFs respectively from the study by Öhlund et al. (2017). The Squamous-PDA Identity signature has been previously described (Somerville et al., 2018). RNA-seq datasets following p63 expression in SUIT2, PATU8988S, HPAFII, AsPC1 and CFPAC1 cells as well as p63 knockout or knockdown in BxPC3 and hF3 cells were from a previous study (Somerville et al., 2018).

## ChIP-Seq library construction and analysis

ChIP procedures were performed as previously described (Somerville et al., 2018). The antibody used for ChIP-seq in this study was H3K27ac (ab4729; abcam). ChIP-seq libraries were constructed using Illumina TruSeq ChIP Sample Prep kit following manufacture's protocol. Briefly, ChIP DNA was end repaired, followed by A-tailing and size selection (300–500 bp) by gel electrophoresis using a 2% gel. 15 PCR cycles were used for final library amplification which was analyzed on a Bioanalyzer using a high sensitivity DNA chip (Agilent). ChIP-seq libraries were single-end sequenced for 50 bp using an Illumina NextSeq platform (Cold Spring Harbor Genome Center, Woodbury). Single end 50 bp sequencing reads were mapped to the hg19 genome using Bowtie2 with default settings (Langmead and Salzberg, 2012). After removing duplicated mapped reads using SAMtools 1.4 (Li et al., 2009), MACS 1.4.2 was used to call peaks using input genomic DNA as control (Feng et al., 2012). Only peaks enriched greater than or equal to 10-fold over input samples were used for subsequent analyses. Metagene plots were made by centering H3K27ac ChIP-seq regions on previously defined Squamous Elements (Somerville et al., 2018) extended to ±10,000 bp with 100 bp bins. Super enhancer analysis was performed using Rank Ordering of Super-Enhancers (ROSE) as described (Lovén et al., 2013; Whyte et al., 2013). p63 ChIP-seq in BxPC3 cells and H3K27ac ChIP-seq in hN34, hN35, hT85, PATU8988S, AsPC1, HPAFII, SUIT2, hF3 and BxPC3, as well as H3K27ac in SUIT2 or BxPC3 cells following p63 expression or knockout were from a previous study (Somerville et al., 2018).

## Single cell RNA-Seq analysis

Gene expression matrices along with sample and cluster annotations of 24 PDAC and 11 normal pancreas samples from the study by Peng et al. (2019) were downloaded from the Chinese National Genomics Data Center (Genome Sequence Archive accession #CRA001160). Raw digital gene expression counts from the aggregated samples were used to visualize expression of genes of interest between sample types (Tumor vs Normal) and across annotated clusters. Data processing to generate an annotated UMAP projection was handled with the Scanpy v1.4.5 software package (Wolf et al., 2018). Briefly, the 35 gene expression matrices were concatenated, normalized, log-transformed and filtered of mitochondrial genes and ribosomal genes prior to principal component analysis, which was restricted to the top 3500 highly variable genes. The top 50 principal components were used to compute the KNN graph and 2D UMAP projection using the default settings. Cells were colored according to cluster assignments from Peng et al. (2019).

## Statistics

For graphical representation of data and statistical analysis, GraphPad Prism was used. Statistical analysis was performed as described in the figure legends.

## Ethics

All animal procedures and studies were approved by the Cold Spring Harbor Laboratory Animal Care and Use Committee (IACUC protocol number 19-16-8).

## Acknowledgements

The authors would like to thank the Cold Spring Harbor Cancer Center Support Grant (CCSG) shared resources: Bioinformatics Shared Resource, Next Generation Sequencing Core Facility, R Rubino and J Habel Animal Resources for technical assistance with the mouse transplantation assays,

P Moody and C Viola in the Flow Cytometry Facility, Animal and Tissue Imaging, and the Animal Facility. TDDS was supported by a grant from the State of New York, contract no. C150158. CRV was supported by Pershing Square Sohn Cancer Research Alliance, the Cold Spring Harbor Laboratory and Northwell Health Affiliation, the National Cancer Institute (NCI) 5P01CA013106-Project 4 and 1RO1CA229699, and a Career Development Award from the Pancreatic Cancer Action Network-American Association for Cancer Research (AACR) 16-20-25-VAKO. XYH was supported by a grant from the State of New York, contract no. C150158. This work was supported by the Lustgarten Foundation, where DAT is a distinguished scholar and Director of the Lustgarten Foundation–designated Laboratory of Pancreatic Cancer Research. DAT is also supported by the Cold Spring Harbor Laboratory Association and the National Institutes of Health (NIH 5P30CA45508, 5P50CA101955, P20CA192996, U10CA180944, U01CA210240, U01CA224013, 1R01CA188134, and 1R01CA190092). CRV, DAT, and ME were supported by the Thompson Family Foundation and Simons Foundation. In addition, we are grateful for support from the following: the Human Frontiers Science Program (LT000195/2015 L for GB), EMBO (ALTF 1203–2014 for GB), Deutsche Forschungsgemeinschaft (DFG) Research Fellowship (DA 2249/1–1 for JDP; KL 3228/1–1, for OK).

# Additional information

## Competing interests

David A Tuveson: an advisor to Surface, Leap, and Cygnal and has stock ownership in Surface and Leap. Christopher R Vakoc: has received funding from Boehringer-Ingelheim and is an advisor to KSQ Therapeutics. The other authors declare that no competing interests exist.

## Funding

| Funder | Grant reference number | Author |
|---|---|---|
| New York State Department of Health | C150158 | Tim DD Somerville |
| Pershing Square Foundation | | Christopher R Vakoc |
| National Cancer Institute | 5P01CA013106-Project 4 | Christopher R Vakoc |
| National Cancer Institute | CA229699 | Christopher R Vakoc |
| Pancreatic Cancer Action Network | 16-20-25-VAKO | Christopher R Vakoc |
| Lustgarten Foundation | | David A Tuveson |
| National Cancer Institute | 5P30CA45508 | David A Tuveson |
| National Cancer Institute | 5P50CA101955 | David A Tuveson |
| National Cancer Institute | P20CA192996 | David A Tuveson |
| National Cancer Institute | U10CA180944 | David A Tuveson |
| National Cancer Institute | U01CA210240 | David A Tuveson |
| National Cancer Institute | U01CA224013 | David A Tuveson |
| National Cancer Institute | 1R01CA188134 | David A Tuveson |
| National Cancer Institute | 1R01CA190092 | David A Tuveson |
| The Cold Spring Harbor Laboratory and Northwell Health Affiliation | | Christopher R Vakoc |
| State of New York | C150158 | Xue-Yan He |
| Thompson Family Foundation | | Mikala Egeblad David A Tuveson Christopher R Vakoc |
| Simons Foundation | | Mikala Egeblad David A Tuveson Christopher R Vakoc |

| Human Frontier Science Program | LT000195/2015 L | Giulia Biffi |
|---|---|---|
| EMBO | ALTF 1203–2014 | Giulia Biffi |
| Deutsche Forschungsgemeinschaft | DA 2249/1–1 | Juliane Daßler-Plenker |
| Deutsche Forschungsgemeinschaft | KL 3228/1–1 | Olaf Klingbeil |

The funders had no role in study design, data collection and interpretation, or the decision to submit the work for publication.

## Author contributions

Tim DD Somerville, Conceptualization, Data curation, Investigation, Methodology; Giulia Biffi, Juliane Daßler-Plenker, Koji Miyabayashi, Yali Xu, Diogo Maia-Silva, Olaf Klingbeil, Osama E Demerdash, Investigation, Methodology; Stella K Hur, Xue-Yan He, Krysten E Vance, Investigation; Jonathan B Preall, Formal analysis; Michael A Hollingsworth, Supervision; Mikala Egeblad, Supervision, Project administration; David A Tuveson, Supervision, Funding acquisition, Project administration; Christopher R Vakoc, Conceptualization, Supervision, Funding acquisition, Project administration

## Author ORCIDs

Tim DD Somerville (iD) https://orcid.org/0000-0002-6295-6126
Christopher R Vakoc (iD) https://orcid.org/0000-0002-1158-7180

## Ethics

Animal experimentation: All animal procedures and studies were approved by the Cold Spring Harbor Laboratory Animal Care and Use Committee (IACUC protocol number 19-16-8).

## Decision letter and Author response

Decision letter https://doi.org/10.7554/eLife.53381.sa1
Author response https://doi.org/10.7554/eLife.53381.sa2

# Additional files

## Supplementary files

• Supplementary file 1. Genes corresponding to the Group 1 and Group 2 PSC clusters and the ranked gene list of Group 2 versus Group 1 genes used for GSEA. Tab-1 (Group 1 genes): List of gene significantly upregulated in the Group 1 PSC cluster versus the Group 2 PSC cluster. Tab-2 (Group 2 genes): List of genes significantly upregulated in the Group 2 PSC cluster versus the Group 1 PSC cluster. Tab-3 (ranked Group 2 vs Group 1): Genes ranked by their mean $\log_2$ fold change in the Group 2 versus the Group 1 PSC cluster.

• Supplementary file 2. iCAF and myCAF gene signatures. Tab-1 (iCAF gene signature): List of 200 mouse genes corresponding to the iCAF gene signature. Tab-2 (myCAF gene signature): List of 200 mouse genes corresponding to the myCAF gene signature.

• Supplementary file 3. Genes significantly upregulated in the human and mouse compartments of SUIT2-p63 versus SUIT2-empty tumors and ranked gene lists used for GSEA. Tab-1 (Human cancer cells sig UP): List of 633 human genes significantly upregulated in the human cancer cell compartment of SUIT2-p63 xenografts versus SUIT2-empty xenografts. Tab-2 (Mouse stromal cells sig UP): List of 500 mouse genes significantly upregulated in the mouse stromal cell compartment of SUIT2-p63 xenografts versus SUIT2-empty xenografts. Tab-3 (Human ranked TP63 vs empty): Human genes ranked by their mean $\log_2$ fold change in the human cancer cell compartment of SUIT2-p63 xenografts versus SUIT2-empty xenografts. Tab-4 (Mouse ranked TP63 vs empty): Mouse genes ranked by their mean $\log_2$ fold change in the stromal cell compartment of SUIT2-p63 xenografts versus SUIT2-empty xenografts.

• Supplementary file 4. Genes significantly downregulated in each sorted fraction of p63-negative versus p63-positive KLM1 tumors and gene lists used for GSEA. Tab-1 (cancer cell sort sig DOWN): List of 459 human genes significantly down regulated in the FACS-purified human cancer cell compartment of p63 knockout versus p63 positive KLM1 xenografts. Tab-2 (fibroblast sort sig DOWN): List of 396 mouse genes significantly down regulated in the FACS-purified mouse fibroblast compartment of p63 knockout versus p63 positive KLM1 xenografts. Tab-3 (immune sort sig DOWN): List of 463 mouse genes significantly down regulated in the FACS-purified mouse immune cell compartment of p63 knockout versus p63 positive KLM1 xenografts. Tab-4 (ranked cancer sgNEG vs sgTP63): Human genes ranked by their mean $\log_2$ fold change in the FACS-purified human cancer cell compartment of p63 knockout versus p63 positive KLM1 xenografts. Tab-5 (ranked CAFs sgNEG vs sgTP63): Mouse genes ranked by their mean $\log_2$ fold change in the FACS-purified mouse fibroblast compartment of p63 knockout versus p63 positive KLM1 xenografts.

• Supplementary file 5. RT-qPCR primer sequences and sgRNA sequences used in this study. Tab-1 (Mouse RT-qPCR primers): List of mouse RT-qPCR primer sequences used in this study. Tab-2 (Human RT-qPCR primers): List of human RT-qPCR primer sequences used in this study. Tab-3 (sgRNAs): List of sgRNA sequences used in this study.

• Transparent reporting form

### Data availability

The RNA-seq and ChIP-seq data in this study is available in the Gene Expression Omnibus database https://www.ncbi.nlm.nih.gov/geo/ with accession number GSE140484.

The following dataset was generated:

| Author(s) | Year | Dataset title | Dataset URL | Database and Identifier |
|---|---|---|---|---|
| Somerville TDD | 2020 | Squamous trans-differentiation of pancreatic cancer cells promotes stromal inflammation | http://www.ncbi.nlm.nih.gov/geo/query/acc.cgi?acc=GSE140484 | NCBI Gene Expression Omnibus, GSE140484 |

The following previously published datasets were used:

| Author(s) | Year | Dataset title | Dataset URL | Database and Identifier |
|---|---|---|---|---|
| Somerville TDD, Xu Y, Miyabayashi K, Tiriac H, Cleary CR, Maia-Silva D, Milazzo JP, Tuveson DA, Vakoc CR | 2018 | TP63-Mediated Enhancer Reprogramming Drives the Squamous Subtype of Pancreatic Ductal Adenocarcinoma | https://www.ncbi.nlm.nih.gov/geo/query/acc.cgi?acc=GSE115463 | NCBI Gene Expression Omnibus, GSE115463 |
| Moffitt RA, Marayati R, Flate EL, Volmar KE, Loeza SGH, Hoadley KA, Rashid NU, Williams LA, Eaton SC, Chung AH | 2015 | Virtual Microdissection of Pancreatic Ductal Adenocarcinoma Reveals Tumor and Stroma Subtypes | https://www.ncbi.nlm.nih.gov/geo/query/acc.cgi?acc=GSE71729 | NCBI Gene Expression Omnibus, GSE71729 |

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
