## [Decision Letter]

**Acceptance summary:**

A major insight from genomic characterization of pancreatic cancers is the recognition of a subtype with squamous differentiation characteristics that is more aggressive clinically and confers an inferior prognosis. This paper provides novel insight in the molecular drivers of these adenosquamous PDA tumors by demonstrating that p63 activates an inflammatory transcriptional program, primarily mediated by upregulation of IL1A and CXCL1, that leads to enrichment of these tumors with neutrophils and cancer associated fibroblasts. The findings add mechanistic specificity to this subtype of pancreas cancer and point toward potential targeted interventions to block these inflammatory signals.

**Decision letter after peer review:**

Thank you for submitting your article "Squamous trans-differentiation of pancreatic cancer cells promotes stromal inflammation" for consideration by *eLife*. Your article has been reviewed by Päivi Ojala as the Senior Editor, a Reviewing Editor, and two reviewers. The following individuals involved in review of your submission have agreed to reveal their identity: Trudy Oliver (Reviewer #1).

The reviewers have discussed the reviews with one another and the Reviewing Editor has drafted this decision to help you prepare a revised submission. While there is enthusiasm for the work, concerns were raised about novelty, in light of two recent related papers from your groups, and about the functional evidence of the iCAFs and neutrophil infiltrates.

Summary:

This work provides new insights into the phenotypic heterogeneity of pancreatic cancer, indicating that p63, which supports a squamous differentiation program, also stimulates an inflammatory microenvironment. The results in co-culture studies establish the ability of p63 expression in tumor cells to stimulate an iCAF signature by paracrine signaling including direct transcriptional upregulation of IL1A. in vivo data support this model and suggest an p63-iCAF inflammatory program promotes tumor growth, potentially involving neutrophil recruitment. The manuscript is clearly presented and the findings have interest relating to both cancer mechanisms and potential clinical significance. Moreover, the correlation between p63 and IL1A expression in clinical specimens is compelling.

There are also some limitations. The authors imply that the positive role of p63 in the growth of tumors in their models involves the activation of the inflammatory program, this is not demonstrated (see essential revision #1). There is also a concern about the novelty of the current work, relative to two recent papers from your group on related topics. Sommerville et al., 2018 shows that TP63 is the driver of adenosquamous PDAC. Britti et al., 2019 shows that IL1 from PDAC cells generates iCAFs. We would like you to address why the current paper represents a significant conceptual advance relative to the two earlier stories or simply connects the dots.

Essential revisions:

1) The study implies that the impact of p63 on pancreatic cancer growth observed in the gain and loss of function studies relates in part to iCAF activation by paracrine signaling. Moreover, the in vitro data presented indicate that IL1A/IL1R1 signaling in the CAF serves as the iCAF switch. Studies to determine the impact of IL1A deletion in tumor cells on in vivo tumor growth, iCAF activation and changes in immune populations would support the significance of the findings. We recognize that such a study (IL1A deletion), if newly initiated, will take some time but would like to hear whether you already have or could generate additional findings on the role of CAFs or neutrophils in vivo. Meta-analysis of the correlation between p63 and the immunological and CAF signatures in public RNAseq datasets for PDA could also strengthen the finding. At a minimum, you should clearly state that the pro-tumor effects observed in vivo could relate to cell autonomous growth regulation rather than any effects on tumor-stroma interplay.

2) It would be more compelling that neutrophils are impacted by p63 expression in Figure 4 if the authors could stain tumors for LY6G (and/or CD11B/MPO) to better confirm that neutrophils are altered in vivo rather than rely on gene expression signatures.

3) The main outstanding question in my opinion is whether the neutrophil recruitment phenotype is truly dependent on CAFs, or whether there are tumor-cell-autonomous mechanisms of neutrophil recruitment. Figure 4 speaks to whether CAFs are a critical intermediate to neutrophil recruitment, but this is only done with one cell line. I wonder whether p63 is sufficient to recruit TANs without CAFs in other cell lines? In Figure 4L, it is not clear whether the increase in neutrophil migration with p63 alone is significant, and why unstimulated PSCs seem to promote neutrophil migration on their own? Related to the main question, there may be tumor-cell-autonomous mechanisms of recruiting TANs via p63-induced chemokines that they have identified like IL1A, IL1B, and CXCL1. I would suggest to move the heatmap of gene expression for the cancer cells from Figure S4C to the main Figure 4 to parallel Figure 4G, 4I. In Figure S4C, it is worth noting that CXCL1 is also reduced in tumor cells, which may directly impact neutrophils in a CAF-independent manner.

---

## [Author Response]

[…] There are also some limitations. The authors imply that the positive role of p63 in the growth of tumors in their models involves the activation of the inflammatory program, this is not demonstrated (see essential revision #1). There is also a concern about the novelty of the current work, relative to two recent papers from your group on related topics. Sommerville et al., 2018 shows that TP63 is the driver of adenosquamous PDAC. Britti et al., 2019 shows that IL1 from PDAC cells generates iCAFs. We would like you to address why the current paper represents a significant conceptual advance relative to the two earlier stories or simply connects the dots.

Unlike other forms of cancer (e.g. lymphoma), it is only in recent years that a consensus has been reached that molecular subtypes exist in pancreatic cancer (Moffitt et al., 2015; Bailey et al., 2016; The Cancer Genome Atlas Research Network, 2017). From these descriptive studies of clinical samples, a major discovery was made that PDA tumors possessing squamous characteristics are more aggressive and portend an inferior prognosis than classical PDA. It is this novel clinical correlation that motivated our study of driver mechanisms of adenosquamous PDA tumors, as well an intense interest in the PDA field with respect to this novel disease entity. Naturally, an emerging focus in the field is in developing personalized therapeutics that exploit the unique properties of these molecular PDA subtypes.

To date, no prior study has demonstrated that adenosquamous PDA possesses enhanced stromal inflammation relative to classical PDA tumors. Importantly, we support this conclusion using both a detailed analysis of clinical PDA specimens (please see below how we have significantly expanded this analysis in our revised manuscript), as well as experimental results using murine and human PDA cell lines and organoid models, ex vivo and in vivo, employing gain- and loss-of function experiments. While this new discovery builds upon our prior work on p63 and iCAFs, it was not obvious *a priori* that adenosquamous PDA would exhibit enhanced stromal inflammation relative to classical PDA. Since inflammation acts as an ‘accelerator’ of PDA progression (Guerra et al., 2007), we believe that our current study will represent a significant milestone in demystifying adenosquamous PDA and leads to non-obvious clinical hypotheses for how treatments might be developed to interrupt inflammatory signaling in this subset of patients. For these reasons, we believe the conceptual advance of our study extends beyond simply drawing mechanistic connections between p63, cytokines, and inflammatory cells, but will motivate a different way of thinking about, and potentially treating, a recently identified form of cancer.

Essential revisions:1) The study implies that the impact of p63 on pancreatic cancer growth observed in the gain and loss of function studies relates in part to iCAF activation by paracrine signaling. Moreover, the in vitro data presented indicate that IL1A/IL1R1 signaling in the CAF serves as the iCAF switch. Studies to determine the impact of IL1A deletion in tumor cells on in vivo tumor growth, iCAF activation and changes in immune populations would support the significance of the findings. We recognize that such a study (IL1A deletion), if newly initiated, will take some time but would like to hear whether you already have or could generate additional findings on the role of CAFs or neutrophils in vivo. Meta-analysis of the correlation between p63 and the immunological and CAF signatures in public RNAseq datasets for PDA could also strengthen the finding. At a minimum, you should clearly state that the pro-tumor effects observed in vivo could relate to cell autonomous growth regulation rather than any effects on tumor-stroma interplay.

In order to address these important points, our revised manuscript includes three new analyses of human PDA specimens or organoids to strengthen the conclusion that adenosquamous tumors are more inflamed than their classical counterparts. First, we have performed multiplex immunofluorescent staining of 12 human PDA tissue samples from a Rapid Autopsy Pancreatic Program, in collaboration with Dr. Tony Hollingsworth at the University of Nebraska, who is now a co-author on our study. Consistent with our prior analysis of bulk tumor tissue samples (Somerville et al., 2018), as well as single cell RNA-seq analysis of human PDA tumors (see below), this analysis revealed a heterogeneous expression pattern of p63 across samples analyzed, enabling us to define p63-negative or p63-positive patient samples (Figure 6E-F and Figure 6—figure supplement 1D). We observed a trend of increased neutrophil infiltration within the p63-positive samples, as indicated by MPO staining (p = 0.06) (Figure 6E-F and Figure 6—figure supplement 1D). Iterative staining of the same samples for αSMA (to mark myofibroblasts) and MUC1 (to mark epithelial tumor cells) did not reveal a difference between the two groups in the number of cells staining positive for these two markers. Taken together with the other findings presented in our manuscript, these observations suggest that adenosquamous PDA tumors exhibit increased neutrophil infiltration that is associated with altered fibroblast function, rather than their absolute number.

Second, we are now including a single cell RNA-seq analysis of 24 human PDA tumors and 11 normal pancreas tissues (extracted from the study by Peng et al., 2019). As shown in Figure 6—figure supplement 1A-B, this analysis identified a subset of tumor ductal (epithelial) cells that aberrantly express *TP63*. Consistent with our prior analysis of bulk tumor tissue samples (Somerville et al., 2018), *TP63* expression was found to be heterogeneous within this patient cohort, enabling us to define *TP63*^high^ and *TP63*^low^ patient samples (Figure 6B). We found that *TP63*^high^ patient samples exhibited significantly higher expression of other squamous marker genes (*KRT5*, *KRT6A*, *S100A2* and *SOX2*) in the ductal epithelial compartment when compared *TP63*^low^ patient samples, consistent with the adenosquamous phenotype (Figure 6C). Importantly, this analysis now allowed us to interrogate gene expression within the fibroblast compartment of these same tumors. Importantly, we observed significantly higher expression of inflammatory genes (e.g. *CXCL1*, *IL6* and *LIF*) in fibroblasts from the *TP63*^high^ group when compared to fibroblasts in *TP63*^low^ tumors (Figure 6D). Unfortunately, this dataset did not contain detectable neutrophil markers in any samples analyzed, suggesting an artifact of sample processing. Nevertheless, this new analysis of clinical specimens at single cell resolution lends strong support for the central conclusion of our study. Of note, we also observed significantly higher, albeit more modest, expression levels of myofibroblast genes (*ACTA2* and *CTGF*) within fibroblasts and stellate cells from the *TP63*^high^ group (Figure 6—figure supplement 6C). The latter observation is in contrast to our findings in vitro and in our NSG xenograft models, which likely reflects the highly complex and dynamic nature of human tumor samples that is not recapitulated in these model systems. For this reason, we have de-emphasized any implication in our study that adenosquamous PDA tumors are depleted of myofibroblast-like cells. Together with the analysis of neutrophils (above, and see below), our revised manuscript emphasizes the acquisition of inflammatory changes within both of these stromal compartments.

Third, we have used early passage organoid cultures of human PDA tumors to analyze the stromal compartment of these models when propagated as an orthotopic xenograft in NSG mice. For this analysis, we compared two human pancreatic tumor organoids classified as either squamous (hF3) or classical (hF27) in our previous study (Tiriac et al., 2018). RNA-seq analysis of the bulk tumor xenograft tissue was performed, followed by mapping of reads to either human or mouse transcriptomes, which confirmed high level *TP63* expression within the hF3 organoid tumor, and the absence of *TP63* expression with the hF27 organoid tumor (Figure 3—figure supplement 1Q). In addition, *TP63* expression in the human tumor compartment was strongly associated with *IL1A* expression and a squamous PDA transcriptional signature (Figure 3—figure supplement 1Q-R). We next interrogated the iCAF signature within the mouse stromal compartments of these two tumors and, consistent with the main conclusion of our study, we observed a significant positive enrichment associated with the hF3 (squamous) tumor versus the hF27 (classical) tumor (Figure 3—figure supplement 1S). Finally, we analyzed the mouse stromal compartments for expression of marker genes highly expressed in neutrophils (*Cxcr2*, *Mpo*, *S100a8* and *S100a9*) and observed high-level expression of these genes within the hF3 stromal compartment, whereas these genes were absent or expressed at very low levels within the hF27 stromal compartment (Figure 3—figure supplement 1T). These observations are in accord with the analysis of primary PDA specimens above, as well as the experimental observations using squamous versus non-squamous PDA cell line xenografts.

It should be emphasized that p63 promotes the expression of several pro-inflammatory cytokines, such as IL-1α and CXCL1. As described below, our revised manuscript provides new evidence that CXCL1 secretion by adenosquamous PDA cells can directly promote neutrophil recruitment (Figure 4—figure supplement 1K-M), which complements results in our original manuscript showing how IL-1α can promote inflammatory changes in fibroblasts/stellate cells. The strong potential for redundancy in this pro-inflammatory response, together with the short time-frame with which we needed to prepare this revision, precluded us from addressing the effects of inactivating IL-1α and/or CXCL1 in PDA xenografts. However, prior studies, including our own, have demonstrated a causal role for IL-1α and CXCL1 in promoting PDA progression in mouse models (Biffi et al., 2019; Li et al., 2018). Additionally, our previous study demonstrated that *Il1a* KO in murine PDA organoids downregulates iCAF formation and impairs tumor growth in vivo (Biffi et al., 2019). We have included additional text in the Discussion to point out how tumor cell autonomous mechanisms also exist downstream of p63 to promote tumor aggressiveness in adenosquamous PDA.

2) It would be more compelling that neutrophils are impacted by p63 expression in Figure 4 if the authors could stain tumors for LY6G (and/or CD11B/MPO) to better confirm that neutrophils are altered in vivo rather than rely on gene expression signatures.

To address this point, we have now performed immunofluorescence staining for LY6G and MPO on the tumor samples harvested from the experiment shown in Figure 4 of the manuscript. As shown in Figure 4—figure supplement 1E-H, p63-negative tumors (sg*TP63*#4) displayed markedly reduced expression of both markers versus the p63-positive (sgNEG) tumors. These data complement our RNA-seq analysis of the stromal compartment of these tumors and demonstrate that the loss of p63 is associated with a significantly reduced neutrophil infiltration phenotype.

3) The main outstanding question in my opinion is whether the neutrophil recruitment phenotype is truly dependent on CAFs, or whether there are tumor-cell-autonomous mechanisms of neutrophil recruitment. Figure 4 speaks to whether CAFs are a critical intermediate to neutrophil recruitment, but this is only done with one cell line. I wonder whether p63 is sufficient to recruit TANs without CAFs in other cell lines? In Figure 4L, it is not clear whether the increase in neutrophil migration with p63 alone is significant, and why unstimulated PSCs seem to promote neutrophil migration on their own? Related to the main question, there may be tumor-cell-autonomous mechanisms of recruiting TANs via p63-induced chemokines that they have identified like IL1A, IL1B, and CXCL1. I would suggest to move the heatmap of gene expression for the cancer cells from Figure S4C to the main Figure 4 to parallel Figure 4G, 4I. In Figure S4C, it is worth noting that CXCL1 is also reduced in tumor cells, which may directly impact neutrophils in a CAF-independent manner.

In order to investigate more deeply the mechanisms of neutrophil recruitment that are driven by p63 expression in PDA cells independently of fibroblasts, we have now performed gain and loss of function experiments using three independent PDA cell lines (Figure 4—figure supplement 1K-M). For these experiments, conditioned media was harvested from SUIT2 and AsPC1 PDA cells ectopically expressing p63, or from KLM1-Cas9 PDA cells infected with an sgRNA targeting *TP63* (sg*TP63*#4), or their respective controls. Conditioned media was then used for neutrophil migration assays as described in Figure 4 of the manuscript (without the addition of fibroblasts/stellate cells). As an additional control, we set up these assays in the presence of a CXCL1 blocking antibody or IgG control. Consistent with the reviewer’s expectation, we consistently observed an increase in neutrophil migration as a result of ectopic p63 expression in PDA cells without the addition fibroblasts (Figure 4—figure supplement 1K-L). In addition, these effects were largely abolished by the presence of a CXCL1 blocking antibody (Figure 4—figure supplement 1K-L). Similarly, in the loss-of-function experiments, *TP63* knockout resulted in a significant reduction in neutrophil migration versus the control cells (sgNEG), and these effects also appear to be dependent on CXCL1 (Figure 4—figure supplement 1M). These data demonstrate that p63 expression in PDA cells drives tumor cell-autonomous neutrophil recruitment in a CXCL1-dependent manner. We would like to highlight that, in our co-culture experiments shown in Figure 4M of the manuscript, the degree of neutrophil recruitment is markedly increased in the p63-postive PDA cell/stellate cell co-culture context (~4-fold increase versus control and ~2-fold increase versus p63-positive PDA cells alone). In this original experiment, the increase in neutrophil migration stimulated by p63 alone did not reach statistical significance (analyzed by two-way ANOVA across all 28 possible interactions in this experiment). Additionally, the degree of neutrophil stimulation observed in the unstimulated PSC condition was comparable to unconditioned DMEM media (control), indicating that unstimulated PSCs did not promote neutrophil migration on their own. We apologize for our original presentation of this figure being unclear. To address this, we have moved the CXCL1 positive control conditions to the supplement and have indicated the lack of significant interactions as described. We conclude that there is indeed a tumor cell-autonomous mechanism of neutrophil recruitment driven by p63 expression in PDA cells (via CXLC1 secretion) and that this phenotype is exacerbated by the presence of fibroblast/stellate cells (via IL-1-mediated iCAF induction). To clarify this point, as suggested by the reviewer, we have moved the heatmap of gene expression from the supplement to the main figure and have now labeled *CXCL1* as a gene regulated by p63 in the cancer cell compartment (Figure 4G).